# Evidence for a DNA-relay mechanism in ParABS-mediated chromosome segregation

Hoong Chuin Lim[1,2], Ivan Vladimirovich Surovtsev[2,3,5], Bruno Gabriel Beltran[4,5], Fang Huang[6], Jörg Bewersdorf[6,7], Christine Jacobs-Wagner[2,3,5,8]*

[1]Department of Molecular Biophysics and Biochemistry, Yale University, New Haven, United States; [2]Microbial Diversity Institute, Yale University, West Haven, United States; [3]Department of Molecular, Cellular and Developmental Biology, Yale University, New Haven, United States; [4]Department of Mathematics, Louisiana State University, Baton Rouge, United States; [5]Howard Hughes Medical Institute, Yale University, New Haven, United States; [6]Department of Cell Biology, Yale School of Medicine, New Haven, United States; [7]Department of Biomedical Engineering, Yale University, New Haven, United States; [8]Department of Microbial Pathogenesis, Yale School of Medicine, New Haven, United States

**Abstract** The widely conserved ParABS system plays a major role in bacterial chromosome segregation. How the components of this system work together to generate translocation force and directional motion remains uncertain. Here, we combine biochemical approaches, quantitative imaging and mathematical modeling to examine the mechanism by which ParA drives the translocation of the ParB/*parS* partition complex in *Caulobacter crescentus*. Our experiments, together with simulations grounded on experimentally-determined biochemical and cellular parameters, suggest a novel 'DNA-relay' mechanism in which the chromosome plays a mechanical function. In this model, DNA-bound ParA-ATP dimers serve as transient tethers that harness the elastic dynamics of the chromosome to relay the partition complex from one DNA region to another across a ParA-ATP dimer gradient. Since ParA-like proteins are implicated in the partitioning of various cytoplasmic cargos, the conservation of their DNA-binding activity suggests that the DNA-relay mechanism may be a general form of intracellular transport in bacteria.

*For correspondence: christine.jacobs-wagner@yale.edu

## Introduction

Perpetuation of life depends on the faithful segregation of genetic material upon division, such that each progeny inherits a full complement of the parental genome. In eukaryotic cells, chromosome segregation is driven by the microtubule-based mitotic spindle in a process for which there is considerable mechanistic understanding (*McIntosh et al., 2012*). In contrast, much less is known in bacteria. Yet a robust segregation mechanism must be in place to keep up with the fast-paced proliferation of bacterial cells. Such segregation mechanism is also critical for maintaining the precise organization of the chromosome and the reproducible positioning of gene loci over generations (*Reyes-Lamothe et al., 2012*; *Wang et al., 2013*).

Recent studies on various bacteria have shown that chromosome segregation is a multi-step process that initiates with the segregation of the duplicated chromosomal origin regions shortly following their replication (*Shebelut et al., 2010*; *Wang et al., 2013*). How the newly replicated chromosomal origin regions segregate remains poorly understood. In recent years, the ParABS systems, which are conserved among most bacterial species (*Livny et al., 2007*), have been recognized as important active transport systems of chromosomal origin regions in diverse bacteria (*Mohl and Gober, 1997*; *Kim et al., 2000*; *Godfrin-Estevenon et al., 2002*; *Lewis et al., 2002*; *Fogel and Waldor, 2006*;

eLife digest DNA molecules exist in cells as tightly packed structures called chromosomes. During the cell cycle, chromosomes duplicate, and the two copies separate, ready to end up in two separate daughter cells. The process of chromosome separation in cells of higher organisms such as animals and plants is well understood. A protein-based structure called the spindle apparatus guides the separating chromosomes to different ends of the dividing cell. However, how chromosome separation occurs in bacteria is not well understood despite its importance in bacterial multiplication.

The nucleoprotein complex ParABS is critical for separating chromosomes in bacteria. The complex is made up of three parts: parS, a stretch of DNA located at the point where chromosome duplication begins; and two proteins called ParB and ParA. While the molecular players are known, how they work together to separate chromosomes is under debate. One popular suggestion is that ParA forms a spindle-like structure. Alternatively, a diffusion-based mechanism has been proposed, where a gradient of ParA molecules bound to the chromosome interacts with a parS/ParB complex, directing the diffusion of the complex.

Lim et al. used quantitative microscopy to observe the movement of the parS/ParB complex and the spatial distribution of the ParA proteins in a model bacterium. The results were inconsistent with the presence of a spindle-like structure in the cells. A mathematical model describing chromosome movement—based on the number and activity of ParA and ParB found in live cells—also failed to fit with either the spindle or the diffusion theory.

Instead, Lim et al. propose a new model, based on the discovery that chromosomes are elastic. In this 'DNA-relay model', ParA is bound to DNA. When ParB intermittently binds to ParA, it usually catches the ParA-DNA complex when the complex is elastically stretched. The elasticity of the chromosome itself then makes the parS/ParB complex move in the direction where the most ParA molecules are still bound to the chromosome. Lim et al. suggest that similar elastic mechanisms could also be behind more general intracellular transport in bacteria.

Saint-Dic et al., 2006; Lasocki et al., 2007; Jakimowicz et al., 2007a, 2007b; Toro et al., 2008; Bartosik et al., 2009; Donovan et al., 2010; Ptacin et al., 2010; Schofield et al., 2010; Shebelut et al., 2010; Harms et al., 2013; Iniesta, 2014). ParABS systems, first identified on plasmids for their role in stable plasmid inheritance (Austin and Abeles, 1983), consist of three components: the DNA sequence parS, the DNA-binding protein ParB, and the deviant Walker A-type ATPase ParA (Ebersbach and Gerdes, 2005). ParB specifically recognizes parS sequences, which are typically found near the origin of replication of most bacterial chromosomes (Livny et al., 2007). Upon binding to parS, ParB is thought to spread on flanking sequences to form the so-called ParB/parS partition complex (Rodionov et al., 1999; Murray et al., 2006; Breier and Grossman, 2007). ParA dimerizes upon ATP binding, which in turn promotes nonspecific DNA binding (Leonard et al., 2005; Hester and Lutkenhaus, 2007). Various in vitro studies have also observed that ParA-ATP dimers can further assemble into filaments (Barilla et al., 2005; Ebersbach et al., 2006; Barilla et al., 2007; Machon et al., 2007; Ptacin et al., 2010). By itself, ParA has a weak ATPase activity but this activity is generally stimulated by an interaction with ParB (Davis et al., 1992; Easter and Gober, 2002; Leonard et al., 2005; Barilla et al., 2007; Ah-Seng et al., 2009; Scholefield et al., 2011). While these biochemical properties have been documented for many ParABS systems, how they give rise to directional transport remains a hot topic of debate (Howard and Gerdes, 2010; Szardenings et al., 2011; Vecchiarelli et al., 2012).

ParABS-mediated chromosomal segregation has probably been most studied in Caulobacter crescentus where ParA and ParB are essential for viability (Mohl and Gober, 1997). In this bacterium, the single, densely packed circular chromosome spans the entire cell and is spatially arranged such that the ParB/parS partition complex and the nearby replication origin are located at the 'old' cell pole while the replication terminus is localized at the opposite, 'new' pole (Jensen and Shapiro, 1999). Epifluorescence microscopy studies in live cells have shown that prior to DNA replication, ParA forms a cloud-like localization pattern that spans from the new pole to about midcell (Ptacin et al., 2010; Schofield et al., 2010; Shebelut et al., 2010). Replication of the origin region results in two physically separated copies of the ParB/parS complex. The ParB/parS complex closer to the old pole remains there while the other one, upon contact with the edge of the ParA cloud, migrates toward the new

pole in the wake of the receding ParA cloud, as if retraction of ParA was 'pulling' the partition complex (*Ptacin et al., 2010*; *Schofield et al., 2010*; *Shebelut et al., 2010*). This correlated spatial dynamics between ParA and ParB/*parS* is a common characteristic of ParABS systems involved in chromosome or plasmid partitioning (*Ebersbach et al., 2006*; *Fogel and Waldor, 2006*; *Hatano et al., 2007*; *Ringgaard et al., 2009*; *Harms et al., 2013*; *Iniesta, 2014*).

The physical mechanism that underlies this correlated dynamics is generally thought to be analogous to the eukaryotic spindle-based mechanism that segregates chromosomes during mitosis (*Gerdes et al., 2010*). According to this popular spindle-like model, ParA polymerizes into a thin filament bundle upon ATP binding. Depolymerization of ParA filaments through ParB-induced ATP hydrolysis then pulls the ParB/*parS* complex and its associated chromosomal origin region. However, the significance of ParA DNA-binding activity remains unclear, even though this activity is essential for the segregation process based on mutational analysis (*Hester and Lutkenhaus, 2007*; *Castaing et al., 2008*; *Ptacin et al., 2010*; *Schofield et al., 2010*). Recent in vitro studies have proposed an alternative 'Brownian-ratchet' mechanism for the partitioning of P1 and F plasmids (*Vecchiarelli et al., 2010*; *Vecchiarelli et al., 2012*, *2013*, *2014*; *Hwang et al., 2013*). However, it is unclear whether the proposed mechanism can support plasmid translocation under physiological conditions.

Apart from chromosome and plasmid segregation, ParA-like proteins have been implicated in the positioning of other cellular components such as metabolic microcompartments and cytosolic chemotaxis clusters (*Savage et al., 2010*; *Ringgaard et al., 2011*; *Roberts et al., 2012*), highlighting the versatility of the ParABS systems. Interestingly, while the chromosomally encoded *Bacillus subtilis* ParA (Soj) and ParB (Spo0J) orthologs have been implicated in chromosome partitioning in sporulating cells (*Ireton et al., 1994*; *Sharpe and Errington, 1996*; *Wu and Errington, 2003*; *Lee and Grossman, 2006*), they are involved in the regulation of DNA replication in vegetative cells (*Murray and Errington, 2008*). *B. subtilis* Spo0J binds to *parS* sites proximal to the origin of replication similar to ParB orthologs involved in chromosome segregation (*Lin et al., 1997*; *Lin and Grossman, 1998*; *Murray et al., 2006*). Like ParA proteins involved in cargo partitioning, Soj forms ATP-dependent dimers that bind DNA in vitro (*Scholefield et al., 2011*) and display nucleoid-associated localization in cells lacking Spo0J (*Murray and Errington, 2008*). However, in wild-type cells, Soj mostly displays a diffuse distribution in the *B. subtilis* cytoplasm (with a weak accumulation at the Spo0J/*parS* location) (*Murray and Errington, 2008*). This pattern is in stark contrast to the cloud-like localization pattern characteristic of ParA orthologs dedicated to cargo transport. Why ParA localization in *B. subtilis* differs remains enigmatic.

In this study, we combined biochemical and cell biological experiments with computational modeling to investigate how the ParABS system drives chromosomal segregation in *C. crescentus*. Our findings are inconsistent with a spindle-like or Brownian ratchet mechanism. Rather, our experiments and simulations support a novel physical mechanism in which chromosome dynamics and the rate of ATP hydrolysis are critical for efficient and robust ParA-dependent translocation of the ParB/*parS* complex. Our data also suggest that the difference in localization and possibly function between *B. subtilis* Soj and *C. crescentus* ParA can be explained by a biochemical difference in ParB's ability to stimulate ParA ATPase activity.

## Results

Since the behavior of a biological system is defined by the biochemical properties of its components inside cells, our first aim was to determine the relevant biochemical properties of *C. crescentus* ParA and ParB proteins in vitro and to correlate these properties with the in vivo conditions. Our ultimate goal was to obtain a mechanistic understanding in the form of a mathematical model.

### ATP-dependent dimerization of ParA

For the in vitro studies, we purified recombinant *C. crescentus* ParA and ParB proteins from the soluble fraction of *Escherichia coli* lysates (*Figure 1—figure supplement 1*). Obtaining stable preparations of chromosomal ParA proteins is notoriously difficult because of the tendency of these proteins to precipitate (*Howard and Gerdes, 2010*). Through troubleshooting, we found that addition of Mg-ATP to all buffers (including the cell lysis and storage buffers) as well as potassium glutamate (150 mM) after chromatography was crucial to maintain long-term stability of purified *C. crescentus* ParA. Such preparations of ParA were used to investigate the oligomerization state of the protein upon ATP binding by size exclusion chromatography. Prior to chromatography, Mg-ATP was removed from a ParA

aliquot via incubation with EDTA followed by buffer exchange ('Materials and methods'). The resulting sample was then split into two samples: one sample remained without ATP, while Mg-ATP was added to the other sample. In the absence of ATP, ParA eluted from the column as a single peak at an elution volume corresponding to the ParA monomer (~28 kDa) (*Figure 1A*). In the presence of Mg-ATP (2.5 mM), ParA eluted earlier, at an elution volume consistent with a dimeric form (~56 kDa) (*Figure 1A*). Thus, *C. crescentus* ParA dimerizes in the presence of ATP concentrations expected to be found inside bacterial cells (*Bennett et al., 2009*).

The absence of peaks at lower elution volumes (including the void volume) suggests that ParA does not form oligomers larger than dimers upon ATP binding under our experimental conditions (*Figure 1A*). Electron microscopy (EM) analysis of ParA (1 µM) in the presence (or absence) of Mg-ATP revealed no evidence of filament formation (data not shown).

## DNA binding switches ParA to an ATP hydrolysis-competent state

Next, we determined the biochemical conditions that affect the ATPase cycle of *C. crescentus* ParA in vitro using an NADH-coupled ATPase assay (*De La Cruz et al., 2000*). In such an assay, ADP produced by ParA ATPase activity is constantly converted to ATP by a pyruvate kinase, continuously replenishing ATP and thus avoiding substrate depletion and potential product inhibition. We determined the specific ATPase activity by measuring the ADP production rate (in $\mu M\ h^{-1}$) and by dividing the result by the concentration of ParA (in µM). By itself, ParA had a very weak ATPase activity that was below the baseline of our assay ($0.5\ hr^{-1}$).

Like other ParA orthologs, *C. crescentus* ParA is known to have an affinity for DNA (*Easter and Gober, 2002*; *Ptacin et al., 2010*; *Schofield et al., 2010*). Addition of increasing concentrations of non-specific DNA resulted in a dose-dependent increase of ParA ATPase activity in vitro (*Figure 1B*). Fitting a Michaelis–Menten curve to the data gave a maximum activity $k_{cat} = 6.7\ hr^{-1}$ and a half saturating DNA concentration $K_{DNA} = 0.15$ mg/ml. Note that DNA failed to stimulate the ATPase activity of the ParA$_{R195E}$ mutant (*Figure 1C*), which can still dimerize upon ATP binding (*Figure 1A*), but does not bind DNA (*Ptacin et al., 2010*; *Schofield et al., 2010*). This suggests that the DNA-dependent activation of ParA ATPase activity requires direct binding between ParA and DNA. In the presence of near saturating concentration of DNA (1.5 mg/ml), the ATPase activity was linearly dependent on ParA concentration (*Figure 1D*). ATPase activity was dependent on the concentration of ATP (*Figure 1E*), resulting in a $K_M$ of 150 µM.

## A high concentration of ParB is required to stimulate ParA ATPase activity

We estimated the concentration of DNA in *C. crescentus* cells to be about 17 mg/ml ('Materials and methods'), which is above the $K_{DNA}$ value of 0.15 mg/ml. This suggests that the DNA concentration inside cells is sufficient to stimulate ParA ATPase activity. However, this ATPase activity of $6.7\ hr^{-1}$ remains slow relative to the time scale of ParB/*parS* migration towards the distal pole, which occurs in the minute range (*Shebelut et al., 2010*; *Ptacin et al., 2010*; *Schofield et al., 2010*; *Hong and McAdams, 2011*). We therefore anticipated that further stimulation by ParB would be required to obtain significant ATPase activity from ParA. Indeed, ParB (60 µM) further increased ParA ATPase activity by about 10-fold in a DNA-dependent fashion (*Figure 1C*). The increased activity was likely due to an interaction between ParB and ParA (as opposed to a contaminating ATPase activity in the ParB preparation) since substitution of ParB by the ParB$_{L12A}$ mutant (purified under the same conditions as ParB, *Figure 1—figure supplement 1D*), which is unable to interact with ParA (*Ptacin et al., 2010*), failed to stimulate ATPase activity (*Figure 1—figure supplement 2*). The synergistic effect of ParB and DNA on ParA ATPase activity indicates that significant ATP turnover only occurs when ParA, ParB and DNA interact.

In the presence of DNA, ParA ATPase activity depended on ParB concentrations (*Figure 1F*) with a surprisingly high apparent $K_{ParB}$ of 80 µM. Equimolar concentrations (3 µM) of ParA and ParB had virtually no effect on ATP turnover; instead much higher concentrations of ParB than ParA were required to obtain significant ATPase stimulation (*Figure 1F*). This is in marked contrast to the *B. subtilis* ParABS system in which equimolar concentrations (2 µM) of Soj (ParA) and Spo0J (ParB) are sufficient for a 70-fold activation of Soj ATPase activity in the presence of DNA (*Scholefield et al., 2011*).

Inside cells, there are two pools of ParB molecules: the free pool that diffuses in the cytoplasm and the associated pool that is bound to the *parS* region. In *C. crescentus*, the high concentration

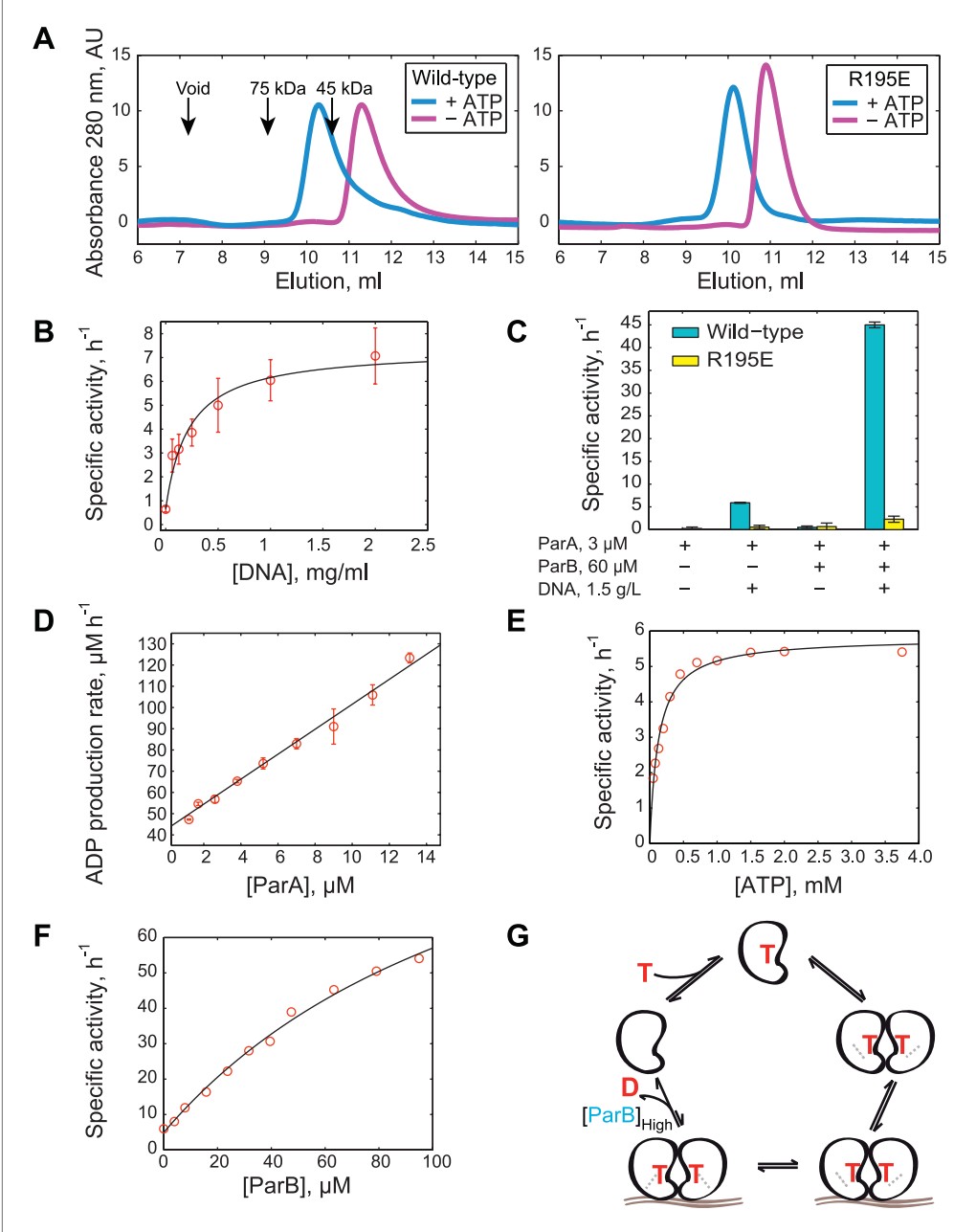

**Figure 1**. Biochemical analysis of ParA ATPase cycle. (**A**) Gel filtration analysis of purified wild-type ParA (*left*) and DNA-binding deficient mutant ParA$_{R195E}$ (*right*) in the presence and absence of ATP. (**B**) Specific activity of ParA (concentration fixed at 6 µM) measured as a function of DNA concentration (0–2.0 mg/ml). Michaelis–Menten equation fit to the data (black) gives $K_{DNA}$ = 0.15 mg/ml. (**C**) Effects of DNA and ParB on ATPase activities of ParA (cyan) and ParA$_{R195E}$ (yellow). (**D**) ADP production rate as a function of ParA concentration measured with fixed ATP (2.5 mM) and DNA (1.5 mg/ml) concentrations. Linear fit to the data (black line) gives $k_{cat}$ = 5.8 hr$^{-1}$. (**E**) Dependence of ATPase activity of ParA upon ATP concentration measured with fixed ParA (10 µM) and DNA (1.5 mg/ml) concentrations. Fitting the Michaelis–Menten equation (black line) gives $k_{cat}$ = 5.9 hr$^{-1}$ and $K_M$ = 150 µM. (**F**) Dependence of ParA (3 µM) ATPase activity on ParB concentrations. Michaelis–Menten equation fit to the data (black line) gives $k_{cat}$ = 120 hr$^{-1}$ and $K_M$ = 80 µM. (**G**) Model of ParA ATPase cycle. ATP-binding promotes ParA dimerization. Upon binding DNA, ParA-ATP dimers presumably adopt an ATP-hydrolysis competent state. A high concentration of ParB is required to stimulate ATP hydrolysis of DNA-associated ParA-ATP dimers. The cycle reinitiates through nucleotide exchange. SDS-PAGE images showing the purity of the purified protein preparations and the results from experiments comparing the stimulatory effects of wild-type (WT) ParB and mutant ParB$_{L12A}$ on

*Figure 1. Continued on next page*

*Figure 1. Continued*

ParA ATPase activity are presented in *Figure 1—figure supplements 1 and 2*. Experiments performed to correlate the biochemical activities measured in vitro with in vivo conditions are presented in *Figure 1—figure supplements 3–8*.

The following figure supplements are available for figure 1:

**Figure supplement 1**. SDS-PAGE analysis of purified protein preparations used in this study.

**Figure supplement 2**. L12A mutation, which prevents ParA–ParB interaction, severely compromises ParB ability to activate ParA ATPase activity.

**Figure supplement 3**. Measurement of GFP-ParB distribution at the subcellular level.

**Figure supplement 4**. Quantification of ParB and ParA abundance in cell lysates by quantitative Western blotting.

**Figure supplement 5**. $His_6$-ParB forms a dimer in solution.

**Figure supplement 6**. Fluorescence-based quantification of ParA-YFP abundance in *C. crescentus* cells.

**Figure supplement 7**. Estimation of the relative abundance of ParA and ParB in vivo.

**Figure supplement 8**. Maximum lengths of different possible ParA polymer configurations.

---

of ParB required for substantial ParA ATPase rate may restrict the ParB-dependent stimulation of ATP hydrolysis to the ParB molecules associated with the *parS* locus. To test this idea, we measured the concentration of *parS*-associated ParB and diffusing ParB inside cells using a strain (MT174) in which native *parB* has been substituted by a functional *parB-gfp* fusion (*Thanbichler and Shapiro, 2006*). Quantitative analysis of the GFP-ParB fluorescent signal ('Materials and methods') showed that GFP-ParB/*parS* foci contain ~80% of the total GFP-ParB signal in the cells while the remaining ~20% correspond to diffusing GFP-ParB plus a potential degradation product (*Figure 1—figure supplement 3A–D*). This distribution of GFP-ParB signal does not vary with cell lengths (*Figure 1—figure supplement 3E*). Using quantitative Western blotting ('Materials and methods', *Figure 1—figure supplement 1C*, *Figure 1—figure supplement 4A*), we determined the total amount of ParB to be about 720 ± 80 molecules/cell (*Table 1*), or ~360 dimers/cell since *C. crescentus* ParB dimerizes in solution (*Figure 1— figure supplement 5*).

The *C. crescentus* chromosome carries two *parS* sites separated by 42 nucleotides (*Livny et al., 2007*; *Toro et al., 2008*). Each site has the potential of binding two ParB molecules. Yet, our measurements indicate that about 580 molecules of ParB (80% of the total 720 ParB molecules) are associated to the *parS* DNA regions (*Table 1*). This is consistent with the notion that upon binding to a *parS* site, ParB spreads to flanking DNA regions to form a large ParB/*parS* assembly (*Rodionov and Yarmolinsky, 2004*; *Murray et al., 2006*; *Breier and Grossman, 2007*). Assuming that the ParB/*parS* assembly is localized within a radius of 78 nm (based on super-resolution microscopy results, *Figure 1—figure supplement 3F*), we estimated a local concentration of ParB to be at least 500 µM at the ParB/*parS* complex (*Table 1*). Based on our in vitro measurements (*Figure 1F*), we expect robust activation of ParA ATPase activity by *parS*-associated ParB inside cells and negligible stimulatory effect by the diffusing ParB (~1 µM).

From these results, we conclude that in *C. crescentus*, only the highly concentrated ParB associated with *parS* (i.e., the ParB/*parS* complex) is capable of stimulating ParA ATPase activity, unlike in *B. subtilis* where even the low concentration of diffusing ParB is likely sufficient to stimulate ATP hydrolysis (*Scholefield et al., 2011*). This explains why ParA (Soj) is mostly monomeric and has a diffused localization in vegetative *B. subtilis* cells because monomers cannot bind DNA (*Scholefield et al., 2011*). Conversely, in *C. crescentus*, ParA can accumulate in a DNA-bound ParA-ATP dimeric form away from the partition complex. Thus, the difference in ParB concentrations required for stimulation of ParA ATPase activity provides an explanation for the difference in ParA localization and possibly function between *B. subtilis* and *C. crescentus* ('Discussion').

**Table 1.** Estimation of ParA and ParB concentrations inside cells

| | Molecules per cell | Mole x 10⁻²¹ | Volume (fL) | Concentration (µM) |
|---|---|---|---|---|
| ParB | 720 ± 80 | 1.2 | 0.25 | 1.8 |
| ParB (diffusing) | 140 | 0.2 | 0.25 | 0.8 |
| ParB (*parS*-associated) | 580 | 1 | 0.002 | 500 |
| ParA | 180 ± 30 | 0.3 | 0.25 | 1.2 |

This table summarizes the abundance and concentrations of ParA and ParB in swarmer/early stalked *C. crescentus* cells. Subcellular distribution of ParB was determined by quantitative fluorescence measurements in cells expressing GFP-ParB (MT174, *Figure 1—figure supplement 3*). ParB abundance was determined by quantitative Western blotting (*Figure 1—figure supplement 4A*). The result showing dimerization of purified ParB in solution is shown in *Figure 1—figure supplement 5*. ParA abundance shown here is the average value determined by three independent methods: (1) Quantitative Western blotting of ParA-YFP abundance in CJW3010 cells in which *parA-yfp* functionally replaces wild-type *parA* in the chromosome (*Figure 1—figure supplement 4B*), (2) Calibrated fluorescence measurement (*Figure 1—figure supplement 6*) and (3) Comparison between ParB-GFP and ParA-YFP amounts by quantitative Western blotting (*Figure 1—figure supplement 7*). The cell volume represents the cytoplasmic volume calculated from a cryoelectron tomograph of a swarmer cell (*Briegel et al., 2008*). The volume occupied by *parS*-associated ParB molecules was inferred from super-resolution PALM images of cells expressing mEos3.2-ParB (CJW4978 strain, *Figure 1—figure supplement 3F*).

## The partition complex does not follow a linear trajectory during ParA-dependent translocation

Apart from the biochemical cycle of ParA (summarized in *Figure 1G*), a key element of the translocation mechanism is the organization of ParA inside the cells. In the prevailing spindle-like model of segregation in *C. crescentus* (*Ptacin et al., 2010*; *Banigan et al., 2011*; *Shtylla and Keener, 2012*), ParA-ATP dimers form a filament bundle that depolymerizes upon interaction with ParB/*parS*, pulling the partition complex toward the new pole. If the partition complex tracks the end of a ParA filament bundle during translocation, we expected that the trajectory of the migrating ParB/*parS* complex would follow a line, reflecting the shape of the underlying ParA filament bundle. So far, the translocation of the partition complex has been characterized along only one cell dimension, the long cell axis (*Shebelut et al., 2010*). Along the cell length, *parS*/ParB motion is known to consist of distinct steps, beginning with a slow, ParA-independent phase during which the two sister partition complexes are separated presumably by 'bulk' segregation mechanisms (such as DNA replication, transcription, entropic unmixing, etc). This is followed by a fast, ParA-dependent phase, and completed by the anchoring of the partition complex at the new pole through a physical interaction between ParB and the PopZ matrix (*Bowman et al., 2008*; *Ebersbach et al., 2008*; *Shebelut et al., 2010*).

To acquire two-dimensional (2D) trajectories (i.e., along both short and long cell axes) at high temporal resolution, we tracked the motion of the partition complex (labeled with GFP-ParB) inside cells at 20-s intervals (*Figure 2A*). Plotting the distance of the segregating complex from the old pole as a function of time for each cell showed the multiphasic movement of the segregating GFP-ParB/*parS* reported before (*Shebelut et al., 2010*), with the sequential slow, fast and anchored phases (*Figure 2B*). We found that the period of the slow, ParA-independent phase increased with faster image acquisition rates (data not shown), indicating sensitivity to phototoxicity. In contrast, the fast, ParA-dependent phase was insensitive to changes in time intervals between image acquisitions (data not shown). We measured that the partition complex covered a distance of 1.0 ± 0.2 µm along the cell length (mean ± standard deviation, SD, n = 141 cells) (*Figure 2C*) within 4.7 ± 1.2 min (*Figure 2D*) during this fast ParA-dependent phase, with an average velocity (run length/run period) of 220 ± 50 nm/min (*Figure 2E*).

Surprisingly, the partition complex did not follow a straight line (or smooth curve) during the fast translocation phase (*Figure 2A*). Moreover, when we examined its motion along the short cell axis, we found that the displacement distributions were virtually identical for the fast and slow phases (*Figure 2F*). These observations are at odds with the idea that a ParA filament bundle serves as a stable bridge guiding the movement of the partitioning complex across the cell. That is, unless the ParA filament bundle rapidly moves across the cell width, which is highly unlikely given the large size of the presumed ParA bundle. We therefore considered two alternative explanations. First, during the

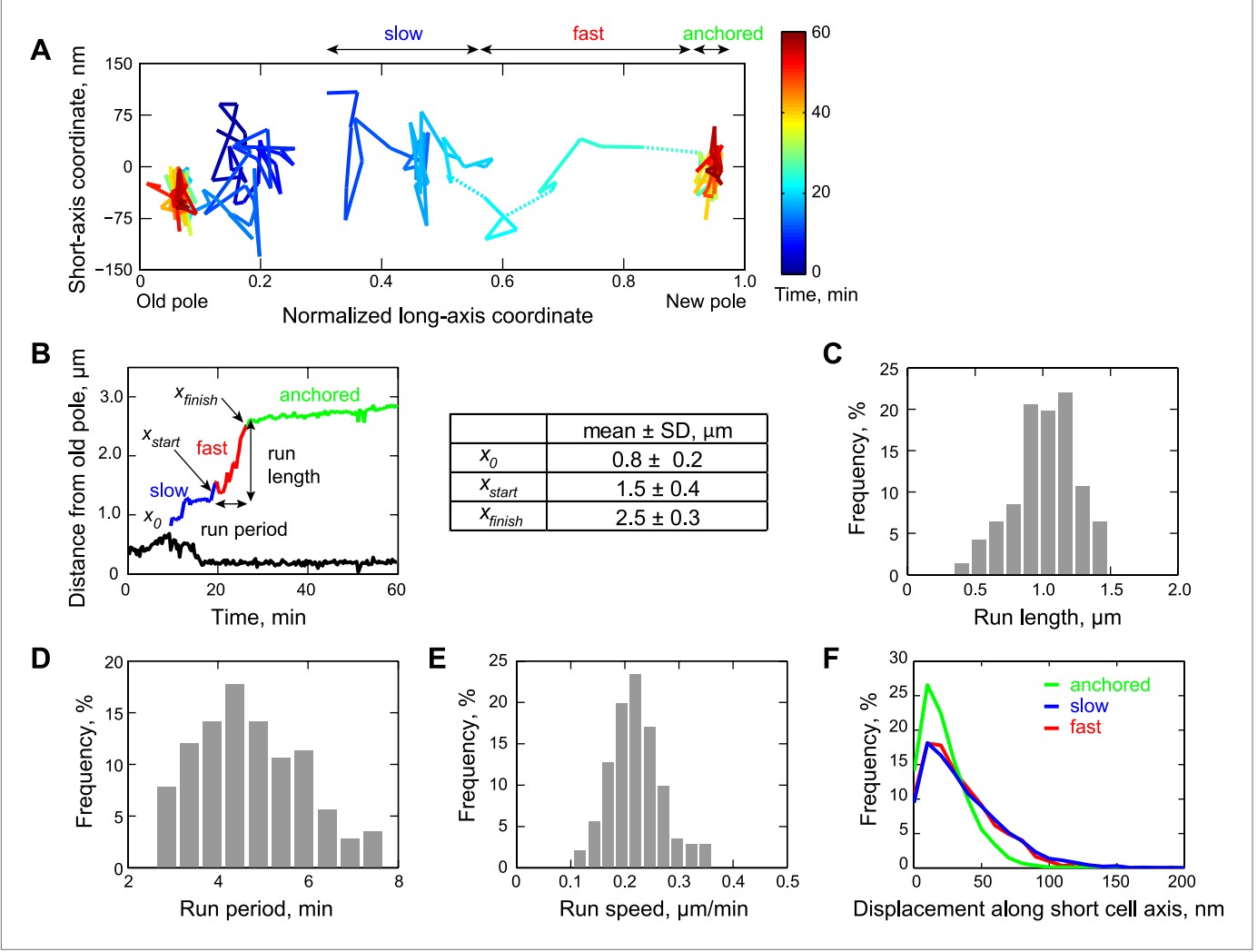

**Figure 2**. Two-dimensional dynamics of the partition complex. (**A**) Representative 2-D trajectory (Δ*t* = 20 s) of duplicated GFP-ParB/*parS* complexes in a CJW4762 cell. Positions of each partition complex detected in consecutive frames were joined with solid lines while dotted lines were used to connect positions interrupted by frame(s) with failed localization. (**B**) Trajectory of GFP-ParB/*parS* complexes in (**A**) along the long cell axis (black, old pole-proximal ParB/*parS* complex; for the distal partition complex: blue, 'slow' phase, red, 'fast' phase, and green, the anchored phase). $x_0$ is the position of the distal partition complex at the moment the second GFP-ParB/*parS* spot appeared in a cell. $x_{finish}$ is the position of the distal partition complex at the moment it became anchored at the new pole. $x_{start}$ is the position when the partition complex transitions from the 'slow' phase into the 'fast' phase. Run length is the distance between $x_{finish}$ and $x_{start}$ along the long cell axis. Run period is the time taken to travel from $x_{start}$ to $x_{finish}$ along the long axis. Also shown is a table summarizing the mean ± SD values for $x_0$, $x_{start}$ and $x_{finish}$. There were more counts for $x_{start}$ and $x_{finish}$ (*n* = 141) than for $x_0$ (*n* = 96) because in some cells the GFP-ParB/*parS* complex had already duplicated prior image acquisition. These values were used to define the start position for the ParB/*parS* complex in our computer simulations and the fast phase in the analysis. (**C**) Histogram showing the distribution of run lengths (*n* = 141). (**D**) Histogram showing the distribution of run periods (*n* = 141). (**E**) Histogram showing the distribution of translocation speeds of partitioning ParB/*parS* complexes during the fast phase (*n* = 141). (**F**) Histogram showing the distribution of short-axis displacements of segregating partition complexes during the slow (blue), fast (red) and pole-anchored (green) phases.

process of translocation, the partitioning complex detaches from the ParA filament bundle and diffuses before reattaching to the ParA filament bundle. Second, ParA-ATP does not form a filamentous structure inside the chromosome-packed cell; instead ParA-ATP dimers bind to the chromosomal DNA away from the ParB/*parS* complexes.

## ParA displays a wide spatial distribution along the cell width

Since the distance covered during the fast, ParA-dependent phase is about 1-µm long (*Figure 2C*), we would expect this distance to approximate the length of the ParA filament bundle if such structure

were involved in the transport mechanism. To examine whether the cellular concentration of ParA could accommodate the formation of a micron-long filament bundle, we determined the number of ParA molecules per cell using three independent methods: quantitative Western blotting (*Figure 1—figure supplement 4B*), calibrated fluorescence microscopy (*Figure 1—figure supplement 6*), and stoichiometric comparison with ParB (*Figure 1—figure supplement 7*). Each method gave a similar result with an average of about 180 ParA molecules or 90 ParA dimers per cell (*Table 1*). ParA2 from *Vibrio cholerae* has been shown to polymerize on DNA in vitro, forming a nucleoprotein filament with a dimension of 4.4 dimer units per helical pitch of 12 nm (*Hui et al., 2010*). If we assume that ParA forms a similar nucleoprotein filament inside *C. crescentus*, this filament would be at most 240 nm in length. Even if we ignore DNA binding and consider that ParA dimers polymerize back-to-back along their largest dimension (as determined from the crystal structure of the *Thermus thermophilus* Soj/ParA dimer; *Leonard et al., 2005*), the filament could be at most 580 nm long (*Figure 1—figure supplement 8*). Thus, the cellular concentration of ParA appears too low to accommodate the formation of even a single micron-long filament.

To gain further insight into the spatial arrangement of ParA, we performed photo-activated localization microscopy (PALM). For this, we generated a functional *parA-dendra2* fusion ('Materials and methods'), which replaced the native *parA* in the chromosome without disrupting the operon structure. In addition, to increase emitter density, we generated a second strain in which ParA-Dendra2 was expressed from a xylose-inducible promoter on the chromosome in addition to being expressed from the native promoter. To specifically examine ParA-Dendra2 localization prior to *parS* replication and segregation, cells in G1 cell cycle phase (swarmer cells) were isolated and imaged by 2D PALM at an averaged localization precision of 18 nm in both the x- and y-directions. Consistent with epifluorescence images (*Ptacin et al., 2010*; *Schofield et al., 2010*; *Shebelut et al., 2010*), ParA-Dendra2 was asymmetrically distributed along the long cell axis (*Figure 3A*). However, we found no evidence of a filament-like localization. Rather, ParA-Dendra2 was widely distributed along the short cell axis (*Figure 3A*). We also imaged cells at a later stage of the cell cycle, and observed strong accumulation of ParA-Dendra2 at the cell pole (*Figure 3A*), consistent with the new-pole localization of ParA accumulation following ParB/*parS* segregation due to ParA's interaction with the polarity factors TipN and PopZ (*Schofield et al., 2010*; *Laloux and Jacobs-Wagner, 2013*).

For comparison, we also imaged crescentin-Dendra2 in *C. crescentus* cells. Crescentin is a bacterial intermediate filament protein that assembles into an inner membrane-associated filamentous structure in *C. crescentus* (*Ausmees et al., 2003*) and is responsible for *C. crescentus*' characteristic crescent-shaped morphology. When crescentin is fused to a fluorescent protein, the crescentin fusion fails to attach to the cell membrane (resulting in loss of cell curvature) but still polymerizes into a single filamentous structure (*Ausmees et al., 2003*). PALM visualization of crescentin-Dendra2 indeed revealed a clear filament-like localization (*Figure 3A*), indicating that our method can identify protein filaments. As a second control, we imaged L1-Dendra2-tagged ribosomes using PALM. In *C. crescentus*, ribosomes spread throughout the cytoplasm, as shown by both cryo-electron tomography and conventional epifluorescence microscopy (*Briegel et al., 2006*; *Montero Llopis et al., 2010*). Super-resolution images of L1-Dendra2-labeled ribosomes were consistent with these observations, showing near homogenous cytoplasmic localization (*Figure 3A*). Importantly, the localization of ParA-Dendra2 (spread along the cell width $\sigma = 146$ nm) was similar to L1-Dendra2-labeled ribosomes ($\sigma = 150$ nm) but not to crescentin-Dendra2 ($\sigma = 32$ nm) (*Figure 3B*). We obtained similar super-resolution images whether we used a commercial N-STORM microscope (see above) or a custom-built microscope (*Figure 3—figure supplement 1*).

Altogether, these results suggest that ParA does not form a thin filamentous structure. Instead, they support a model in which the asymmetric ParA cloud observed by epifluorescence microscopy consists of sparse ParA-ATP dimers (or small oligomers) bound to the chromosome away from the ParB/*parS* complex.

## Modeling suggests that diffusion and ParA/ParB interplay are not sufficient to drive directed translocation

How can the properties we observed lead to a robust directional motion of ParB/*parS* complex toward the new pole? We considered the possibility that ParB/*parS* is simply diffusing and that its binding to the DNA-bound ParA-ATP dimers results in a biased diffusion along the ParA-ATP dimer gradient, as we previously suggested (*Schofield et al., 2010*). This mechanism would be similar to the Brownian ratchet proposed for the P1 and F plasmids (*Hwang et al., 2013*; *Vecchiarelli et al., 2013*, *2014*).

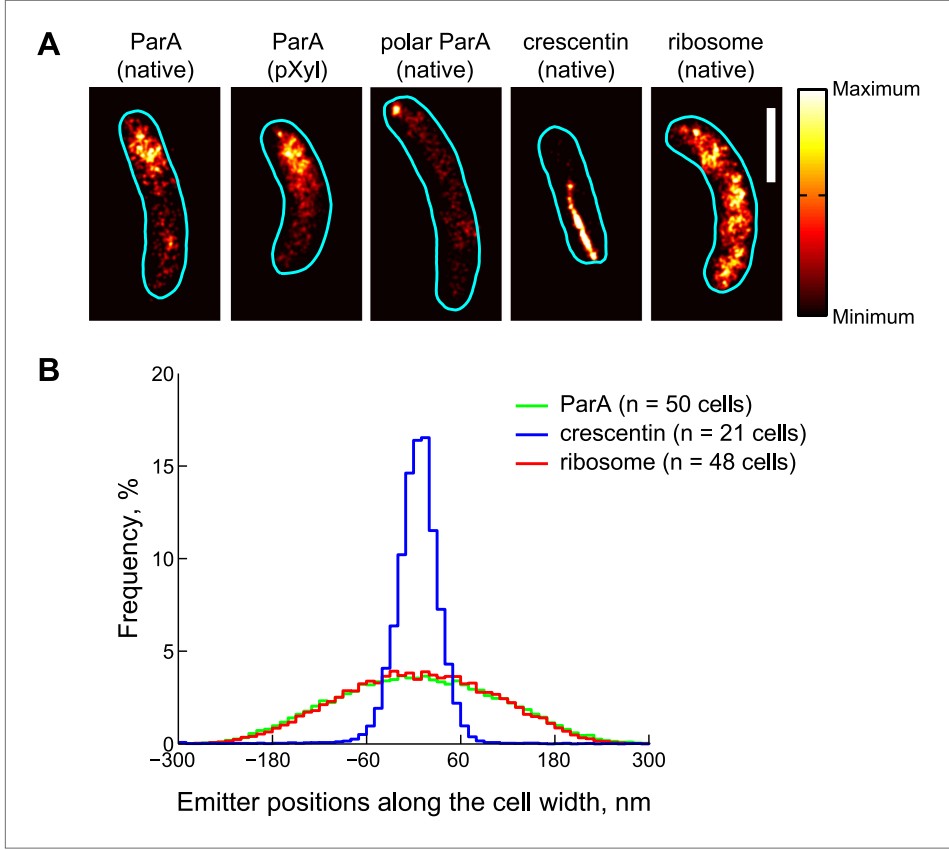

**Figure 3**. Subdiffraction visualization of ParA-Dendra2 in live cells. (**A**) Representative super-resolution PALM images of cells producing ParA-Dendra2 at native levels in place of ParA (strain CJW4915), ParA-Dendra2 at xylose-inducible levels in addition to ParA-Dendra2 expressed from the native promoter (CJW5154), crescentin-Dendra2 at native levels (CJW4902) and ribosome-associated L1-Dendra2 at native levels (CJW5156). Synthesis of ParA-Dendra2 in CJW5154 cells was induced with 0.3% xylose for 1 hr before swarmer cells were isolated for imaging. Scale bar = 1 μm. (**B**) Positional distributions of single emitters along the short-cell axis for cells expressing ParA-Dendra2 (CJW5154, green), crescentin-Dendra2 (CJW4902, blue) and L1-Dendra2 (CJW5156, red). For each cell, a 0.5–0.8 μm segment containing at least 600 emitters was selected. Emitter positions along the short cell axis were calculated relative to the mean position. Additional super-resolution microscopy images and analysis are presented in *Figure 3—figure supplement 1*.

The following figure supplements are available for figure 3:

**Figure supplement 1**. Comparison of ParA and crescentin localization by super-resolution imaging.

In this proposed mechanism (hereafter referred to as 'diffusion-binding' mechanism, *Figure 4A*), the DNA serves as a matrix to tether ParA-ATP dimers and the partition complex diffuses until it binds to one or more DNA-bound ParA-ATP dimers. Stimulation of ATP hydrolysis results in monomerization of ParA, causing the release of the ParB/*parS* complex and a local depletion of DNA-bound ParA-ATP dimers. The partition complex then encounters new DNA-bound ParA-ATP dimers by diffusion, reinitiating the biochemical cycle. The hypothesis is that repeated cycles of this process would result in biased diffusion of ParB/*parS* toward a higher concentration of ParA-ATP dimers.

To test this hypothetical mechanism, we developed a 'diffusion-binding' model ('Materials and methods') that was constrained by experimentally-determined parameters (*Table 2*) to best reflect the in vivo situation. The ParB-rich partition complex was modeled as a sphere with a radius of 50 nm to simulate the interaction radius of the partition complex that we estimated from our super-resolution images (*Figure 1—figure supplement 3*). The ParB sphere was allowed to interact with multiple DNA-bound ParA-ATP dimers at any given time to reflect the high number of ParB molecules (i.e., ParA binding sites) within the partition complex (*Table 1*, *Figure 1—figure supplement 3*). The rate constant of

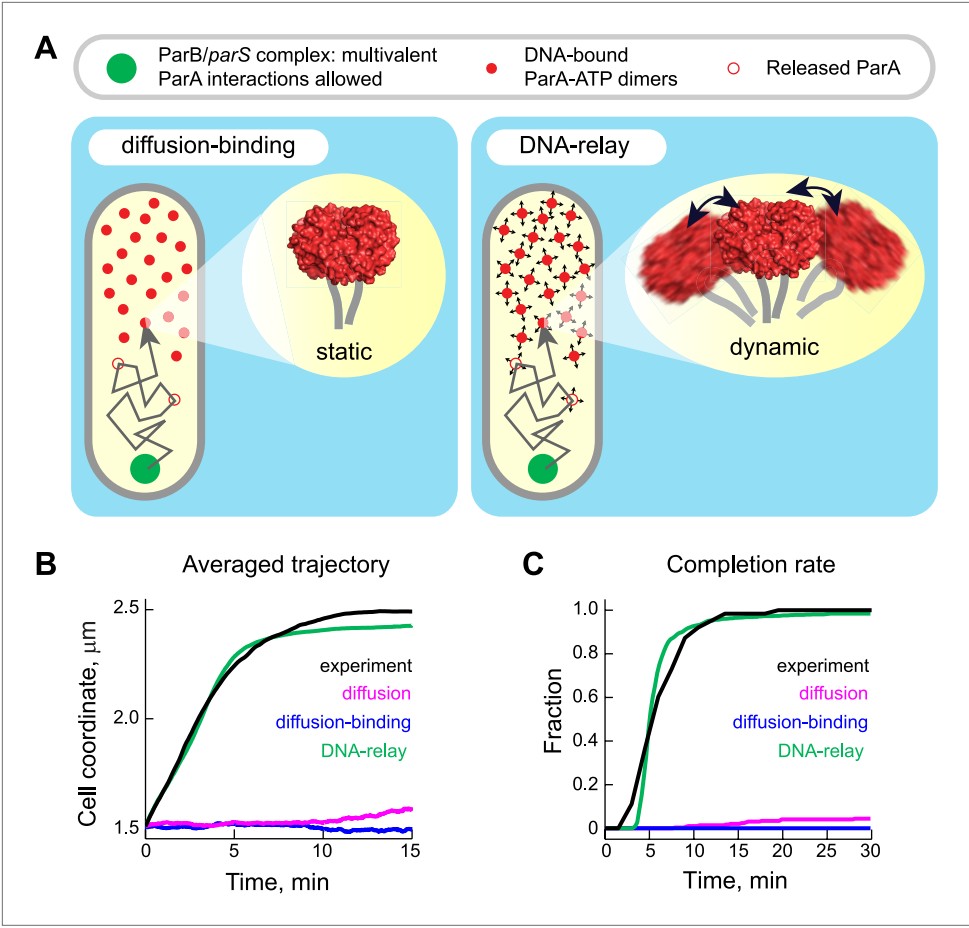

**Figure 4**. The DNA-relay model results in robust translocation of the ParB/*parS* complex. (**A**) *Left*, graphical representation of the 'diffusion-binding' model: the diffusing ParB/*parS* complex (green disk) interacts with DNA-immobilized ParA-ATP dimers (filled red disks), stimulates ATPase activity, resulting in the dissociation of ParA from the DNA (open red disks). *Right*, graphical representation of the 'DNA-relay model': same as in (**A**) except that DNA-bound ParA-ATP dimers fluctuate randomly according to the movement of the associated DNA. When the ParB/*parS* complex is associated with one or more DNA-bound ParA-ATP dimers, it experiences the elastic force governing the dynamics of the DNA loci associated with ParA dimers. (**B**) Averaged positions of the simulated ParB/*parS* complex as a function of time. (**C**) Fraction of trajectories that completed translocation as a function of time. A summary of all parameters used in the simulations is presented in *Table 2*. Additional experiments performed to obtain these parameter values are presented in *Figure 4—figure supplements 1 and 2*.

The following figure supplements are available for figure 4:

**Figure supplement 1**. Average ParA-YFP fluorescence profiles during ParB/parS segregation.

**Figure supplement 2**. Estimation of the diffusion coefficient of the ParB/*parS* complex, $D_{PC}$.

ParB-stimulated ATPase activity was 0.03 s$^{-1}$, as measured in our biochemical study (*Figure 1F*). The concentration of ParA-ATP dimers was 90 per cell (*Table 1*). Their spatial distribution was modeled to reproduce the gradient of ParA-ATP dimers bound to the DNA matrix inside cells. The shape of the ParA gradient was determined by quantitative analysis of cells expressing ParA-YFP at different stages of segregation (*Figure 4—figure supplement 1*). The diffusion coefficient of the partition complex ($D_{PC}$) was estimated to be 0.0001 μm$^2$ s$^{-1}$ from tracking the motion of detached GFP-ParB/*parS* prior to the directed ParA-dependent phase (*Figure 4—figure supplement 2* and 'Materials and methods').

We generated over 1000 trajectories in virtual cells using Brownian dynamics simulation. We focused our analysis of the simulated trajectories to the region between 1.5 μm ($x_{start}$) and 2.5 μm ($x_{finish}$) from the old pole of virtual cells since this 1-μm region corresponds to the active, ParA-dependent motion

**Table 2.** Default parameters and values used in simulations of mathematical models

| Model | Parameter | | Value | Comments | Source |
|---|---|---|---|---|---|
| 1, 2, 3 | Time step of simulations | $\Delta t$ | 0.001 s | No significant differences with simulations using smaller steps | |
| 1, 2, 3 | Number of simulated trajectories | $n_{runs}$ | 1024 | | |
| 1, 2, 3 | Duration of simulations | $t_{fin}$ | 2000 s | ~3 × time scale of translocation | |
| 1, 2, 3 | Cell length | $l_0$ | 2.6 µm | Average cell length and width of cells with first appearance of 2 ParB foci | This study |
| | Cell width | $w_0$ | 0.4 µm | | |
| 1, 2, 3 | Initial coordinates of ParB/*parS* complex, relative to the long cell axis(0 = old pole; $l_0$ = new pole) and short axis (cell walls at $-w_0/2$ and $w_0/2$) | $x_0$ | 0.8 µm | Average coordinate of the distal ParB focus in cells at the first appearance of 2 ParB foci | This study |
| | | $y_0$ | 0.0 µm | | |
| 1, 2, 3 | Start of 'fast' phase (used only in analysis of the simulations) | $x_{start}$ | 1.5 µm | Calculated from $x_{finish}$ - run length (from **Figure 2D**) | This study |
| 1, 2, 3 | End point of translocation | $x_{finish}$ | 2.5 µm | Average coordinate at which distal ParB focus became anchored | This study |
| 1, 2, 3 | Radius of the disk for ParB/*parS* complex | $R_{ParB}$ | 50 nm | Value close to an estimate from super-resolution images | This study |
| 2, 3 | Radius of the disk for ParA dimer | $R_{ParA}$ | 2 nm | Value close to the dimension of the crystal structure of a Soj dimer (PDB: 2BEK) | (**Leonard et al., 2005**) |
| 2, 3 | Number ParA dimers | $n_{ParA}$ | 90 | Average of three measurements by different techniques | This study |
| 2, 3 | Rate of ParB-stimulated hydrolysis of ATP by ParA dimers | $k_{cat}$ | 0.03 s$^{-1}$ | Best fit value to ParB dependence curve | This study |
| 2, 3 | Rate of ParA dimer rebinding to the DNA | $k_{db}$ | 0.03 s$^{-1}$ | Results do not depend on the exact values (0.01–1 s$^{-1}$ range tested) | |
| 2, 3 | Spatial distribution of DNA-bound ParA dimers | $P_{ParA-DNA}$ | **Equation 6** | Measured from ParA-YFP fluorescence profile during segregation | This study |
| 1, 2, 3 | Diffusion coefficient of the translocating ParB/*parS* complex | $D_{PC}$ | 0.0001 µm²s$^{-1}$ | Estimated from non-directional phases of ParB trajectories | This study |
| 3 | Diffusion coefficient of DNA-bound ParA dimers | $D_A$ | 0.01 µm²s$^{-1}$ | Calculated from the time-dependent generalized diffusion coefficient | (**Weber et al., 2010**; **Javer et al., 2013**) |
| 3 | Standard deviations of fluctuating DNA-bound ParA dimers used to define elastic constants ($1/\sigma^2 = k_{sp}/kT$) | $\sigma_{long}$ | 0.06 µm | Measured from the positional fluctuation of the *groESL*, 139_lac and 165_lac DNA loci. | This study |
| | | $\sigma_{short}$ | 0.04 µm | | |

Model 1: diffusion, Model 2: diffusion-binding, Model 3: DNA-relay.

(fast phase) in real cells (**Figure 2B**). We found that the averaged trajectory showed little (if any) net translocation toward the new pole (**Figure 4B**). In fact, none of the simulated trajectories were able to cross this 1-µm region within 30 min (**Figure 4C**; **Video 1** for a representative trajectory). This is in marked contrast to experimental observations where most cells complete translocation of ParB/*parS* within that time frame (**Figure 2D**, **Figure 4C**). Moreover, the diffusion-binding model did not work better than a simple diffusion model ('Materials and methods') in which the partition complex diffuses without interacting with ParA-ATP dimers (**Figure 4B,C**; **Video 2**).

The diffusion-binding model did not lead to directional motion, indicating that diffusion of the ParB/*parS* complex coupled with its biochemical interplay with DNA-bound ParA-ATP dimers does not create a Brownian ratchet. This is because this mechanism is only governed by the diffusion of the partition complex. There is no force involved and nothing prevents the partition complex from

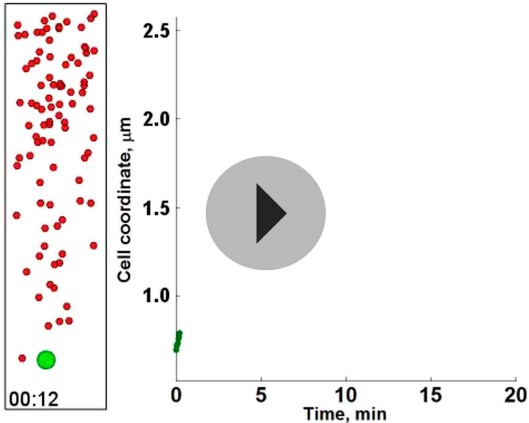

**Video 1**. Example of a simulated trajectory for the diffusion-binding model. Trajectories were generated by Brownian dynamics simulations with time step $dt$ = 1 ms and the following parameters: diffusion coefficient of the partition complex $D_{PC}$ = 0.0001 μm² s⁻¹, total number of ParA dimers $n_{ParA}$ = 90, ParB-stimulated rate of ParA ATPase activity $k_{cat}$ = 0.03 s⁻¹, rate constant for DNA-binding $k_{DB}$ = 0.03 s⁻¹. The partition complex and ParA-ATP dimers are shown as green and red spheres, respectively. Shown here is a representative trajectory in absolute cell coordinate (0 μm = old pole; 2.5 μm = new pole) as a function of time in a virtual cell.

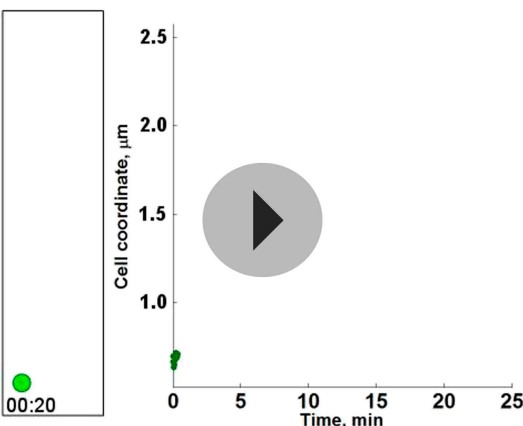

**Video 2**. Example of a simulated trajectory for the diffusion model. Trajectories were generated by Brownian dynamics simulations with time step $dt$ = 1 ms and a diffusion coefficient of partition complex $D_{PC}$ = 0.0001 μm² s⁻¹. The partition complex is shown in green. Shown here is a representative trajectory in absolute cell coordinate (0 μm = old pole; 2.5 μm = new pole) as a function of time in a virtual cell.

diffusing in the wrong direction when the interaction between ParB and ParA is lost following ATP hydrolysis (**Video 2**). The transient interactions with DNA-bound ParA-ATP dimers only intermittently stall the motion of the partition complex. These results suggested that something was missing from the diffusion-binding model.

## Adding chromosome dynamics to the model results in translocation properties similar to those observed in vivo

So far, our diffusion-binding model only considered the chromosome as a static matrix for the attachment of ParA-ATP dimers. Although chromosomal loci keep their average physical position inside bacterial cells (**Viollier et al., 2004**; **Wiggins et al., 2010**), they are mobile and exhibit discernible motion independent of chromosome segregation (**Espeli et al., 2008**; **Weber et al., 2010**; **Hadizadeh Yazdi et al., 2012**; **Javer et al., 2013**). This suggests that the DNA-bound ParA-ATP dimers inside cells are not static; instead, they fluctuate locally due to their associations with a dynamic DNA matrix.

To realistically describe chromosome dynamics in our model, we tracked the dynamics of a LacI-CFP-labeled chromosomal locus (*groESL*) positioned near the middle of the *C. crescentus* cell prior to replication and segregation. We performed time-lapse imaging at 2-s interval and only analyzed cells with a single fluorescent locus to eliminate motions due to segregation. Each locus (n = 641) moved randomly around an equilibrium position within individual cells (**Figure 5A**). To measure temporal fluctuations in position at the single-cell level, we determined the deviations of the locus position from the equilibrium point (i.e., mean position for each trajectory) at each time-point. The distribution of *groESL* position deviations was well approximated by an asymmetric two-dimensional (2-D) Gaussian distribution (**Figure 5B**). This indicates that the DNA locus moves in an asymmetric 2-D harmonic potential, which implies elastic dynamics. This is consistent with the recent proposal that bacterial chromosomes behave like elastic filaments (**Wiggins et al., 2010**).

Gaussian fitting along long and short cell axes yielded standard deviations $\sigma_{long}$ = 0.064 ± 0.002 μm (best fit ± error of fitting) and $\sigma_{short}$ = 0.038 ± 0.001 μm (**Figure 5B**), which were converted (using **Equation 2**, 'Materials and methods') into effective spring constants ($k_{sp}$) of 0.001 pN/nm and 0.003 pN/nm along the long and short axes, respectively. We also measured the fluctuations of two other chromosomal loci (139_lac and 165_lac) at positions 1599540 and 2481399 on the chromosomal map (**Viollier et al., 2004**) and observed similar elastic behaviors (**Figure 5C**). We used

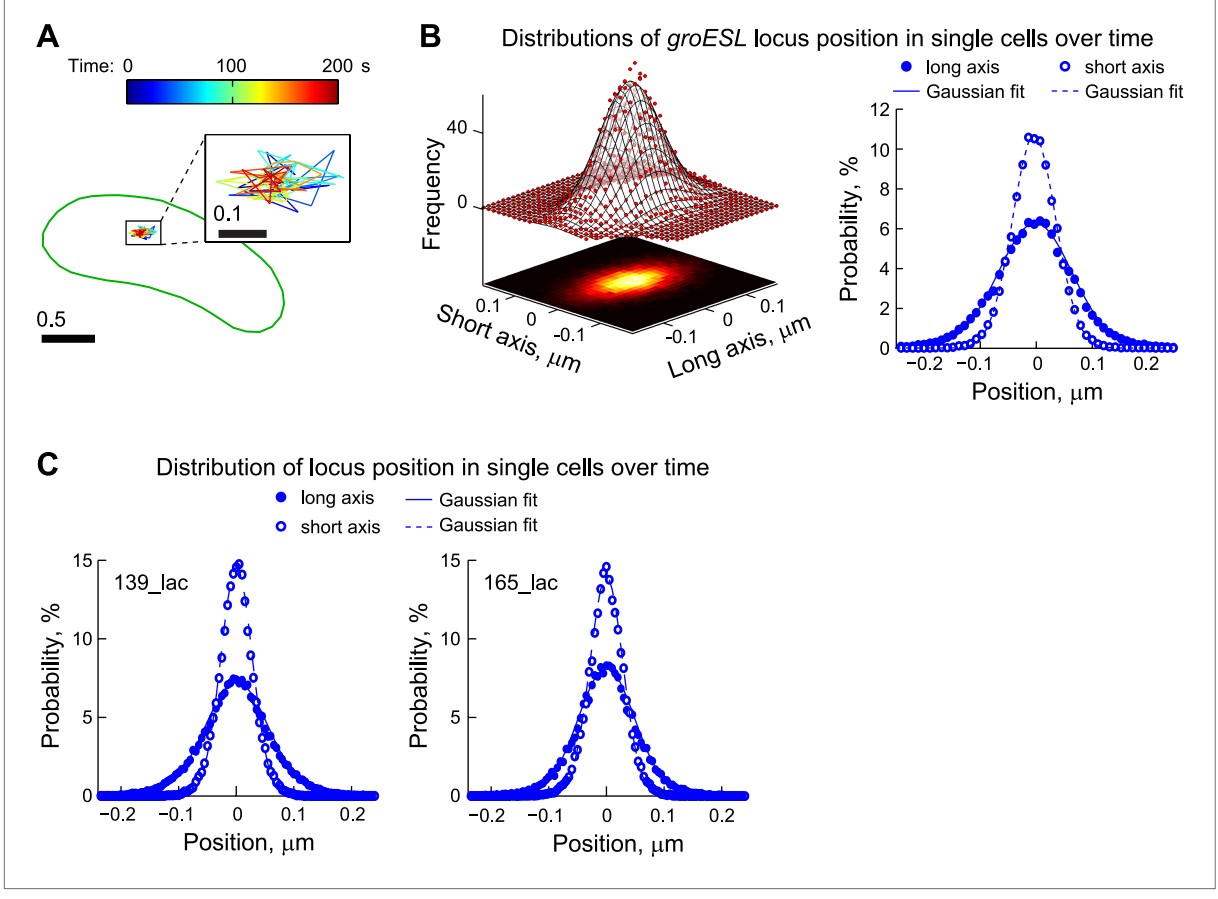

**Figure 5**. Chromosomal loci exhibit elastic dynamics. (**A**) Representative trajectory of a LacI-CFP-labeled *groESL* locus tracked in live CJW2966 cells showing the dynamics of a chromosomal locus. The DNA locus was imaged every 2 s for 180 s. The displacements were colored as a function of time and overlaid with the cell outline (green). The zoomed trajectory in the inset shows the magnitude of displacements. To induce LacI-CFP expression, cells were incubated for 1 hr with 0.03% xylose prior to imaging. (**B**) *Left*, two-dimensional distribution of the *groESL* locus positions relative to the mean position of each trajectory. The scatter plots were generated from locus positions of 641 trajectories while the mesh surface is the best 2-D asymmetrical Gaussian fit (with $\sigma_{long}$ = 0.06 µm and $\sigma_{short}$ = 0.04 µm). The lower plane is a heatmap of the experimental data. *Right*, the same data set is represented as 1-D distributions of the *groESL* locus positions relative to the mean position of each trajectory along the long and short cell axes in single cells. Experimental data (filled circles, long axis; open circles, short axis) and Gaussian fits (solid line, long axis; dashed line, short axis) are shown. (**C**) 1-D distributions of the 139_lac and 165_lac locus positions (**Viollier et al., 2004**) relative to the mean position of each trajectory along the long and short cell axes in single cells. *Left*, asymmetrical Gaussian fit gives $\sigma_{long}$ = 0.06 µm and $\sigma_{short}$ = 0.03 µm for the 139_lac locus in which the *lac* operators are inserted at position 1599540 on the *C. crescentus* chromosome (CJW5466 strain). *Right*, asymmetrical Gaussian fit gives $\sigma_{long}$ = 0.05 µm and $\sigma_{short}$ = 0.03 µm for the 165_lac locus in which the *lac* operators are inserted at position 2481399 on the *C. crescentus* chromosome (CJW5468 strain).

the mean $\sigma_{long}$ and $\sigma_{short}$ values of these three chromosomal loci (**Table 2**) to quantitatively incorporate the intrinsic fluctuating motion of DNA-bound ParA-ATP dimers into the diffusion-binding model. Simulation of this revised model, named the 'DNA-relay' model hereafter (**Figure 4A**), showed that considering the DNA as a dynamic matrix has a dramatic effect, yielding fast and robust translocation of partition complexes (**Figure 4B,C**; **Video 3**).

The ability of the DNA-relay model to reproduce the experimental results (**Figure 4B,C**) is particularly remarkable given that the model was grounded on experimentally measured data (**Table 2**) and did not rely on parameter optimization. In this model, the DNA-bound ParA-ATP dimers fluctuate around the equilibrium point of the underlying DNA loci. When the partition complex catches DNA-bound ParA-ATP dimers in a stretched out-of-equilibrium state, it experiences the elastic force, which moves the complex toward the equilibrium point until ATP hydrolysis releases it. Repetition of the process results in a *relay* of the partition complex from one chromosomal region to another. Directionality arises from the presence of the ParA-ATP dimer gradient. Since there are more DNA-bound ParA-ATP

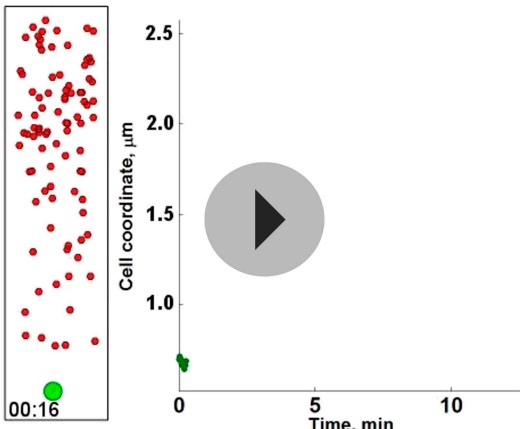

**Video 3**. Example of a simulated trajectory for the DNA-relay model. Trajectories were generated by Brownian dynamics simulations with time step $dt = 1$ ms and following parameters: diffusion coefficient of partition complex $D_{PC} = 0.0001$ µm² s⁻¹, total number of ParA dimers $n_{ParA} = 90$, ParB-stimulated rate of ATPase activity $k_{cat} = 0.03$ s⁻¹, rate constant for DNA binding $k_{DB} = 0.03$ s⁻¹ and spring constant $k_{sp}/kT = 1/\sigma^2 = 280$ µm⁻². The partition complex and ParA-ATP dimers are shown as green and red spheres, respectively. Shown here is a representative trajectory in absolute cell coordinate (0 µm = old pole; 2.5 µm = new pole) as a function of time in a virtual cell.

dimers toward the new pole, the partition complex will tend to encounter these DNA-bound ParA-ATP dimers as they stretch toward the partition complex. This results in a net force and hence translocation relay toward the new pole.

From our estimate of diffusion coefficient of the partition complex ($D_{PC} = 0.0001$ µm² s⁻¹, see *Figure 4—figure supplement 2*) and its velocity during ParA-dependent phase ($v = 0.003$ µm/s, *Figure 2E*), one can estimate that the partition complex experiences a force $F = v\,(kT/D_{PC}) = 0.1$ pN (where $k$ is the Boltzmann constant and $T$ is the absolute temperature). This is in good agreement with the characteristic elastic force generated from chromosomal locus dynamics, which can be estimated as $F = k_{sp}\,\sigma_{long} = 0.06$ pN (where $k_{sp}$ is the measured spring constant and $\sigma_{long}$ is the standard deviation of the DNA locus from equilibrium). Thus, one or more interactions with DNA-bound ParA dimers provide sufficient force to account for the observed partitioning velocity, providing further support for the DNA-relay mechanism.

## The ParB-bound *parS* region transiently decompacts during translocation

The elastic force in the DNA-relay mechanism implies that the partition complex is under tension when it is bound to DNA-associated ParA-ATP dimers (i.e., during ParA-dependent translocation). We found evidence supporting this notion when we examined the fluorescent GFP-ParB/*parS* signal. The GFP-ParB/*parS* signal formed a diffraction-limited focus when located close to the cell poles, that is, before or after segregation when GFP-ParB/*parS* was not interacting with ParA. However, the fluorescent signal associated with the segregating GFP-ParB/*parS* region (i.e., interacting with DNA-bound ParA dimers) frequently adopted an extended configuration (*Figure 6A*), consistent with a force decompacting the *parS* region. We confirmed that this extended conformation was not due to rapid motions of GFP-ParB/*parS* during image acquisition as it was readily observed in formaldehyde-fixed cells (*Figure 6B*).

To quantify these observations, we used a filtering-based algorithm to identify fluorescent objects (CFP-ParB/*parS* complexes) in each cell (*Parry et al., 2014*). Analyzing only cells in which the CFP-ParB/*parS* complex has duplicated and separated, we measured the aspect ratio (AR) of each partition complex (*Figure 6C*), a metric that has also been used to distinguish between compacted and decompacted states of chromatin loci in eukaryotic nuclei (*Verdaasdonk et al., 2013*). A perfectly round CFP-ParB/*parS* signal is expected to have an AR of 1 while a stretched signal has an AR score greater than 1 (*Figure 6C*). We visually correlated a subset of the spots with their AR and verified that an AR cutoff of 1.5 reliably discriminated between the compacted and decompacted conformations. Using this AR cutoff, we found that the frequency of decompacted conformation (AR >1.5) is low near the new and old poles (*Figure 6D*), where the ParB/*parS* complexes are expected to either be anchored by PopZ or diffuse locally (*Figure 2A*). However, the frequency of decompacted CFP-ParB/*parS* signals increased as the partition complex moved away from the old pole (*Figure 6D*). Thus, the frequency of decompacted CFP-ParB/*parS* signals correlated with the concentration of ParA-ATP dimers, suggesting that the stretched conformation adopted by the partition complex is triggered by ParB interacting with ParA-ATP dimers bound to the dynamic DNA. Furthermore, the AR of the segregating partition complex appeared to fluctuate over time (*Figure 6E*), consistent with the partition complex dynamically responding to fluctuating pulling forces.

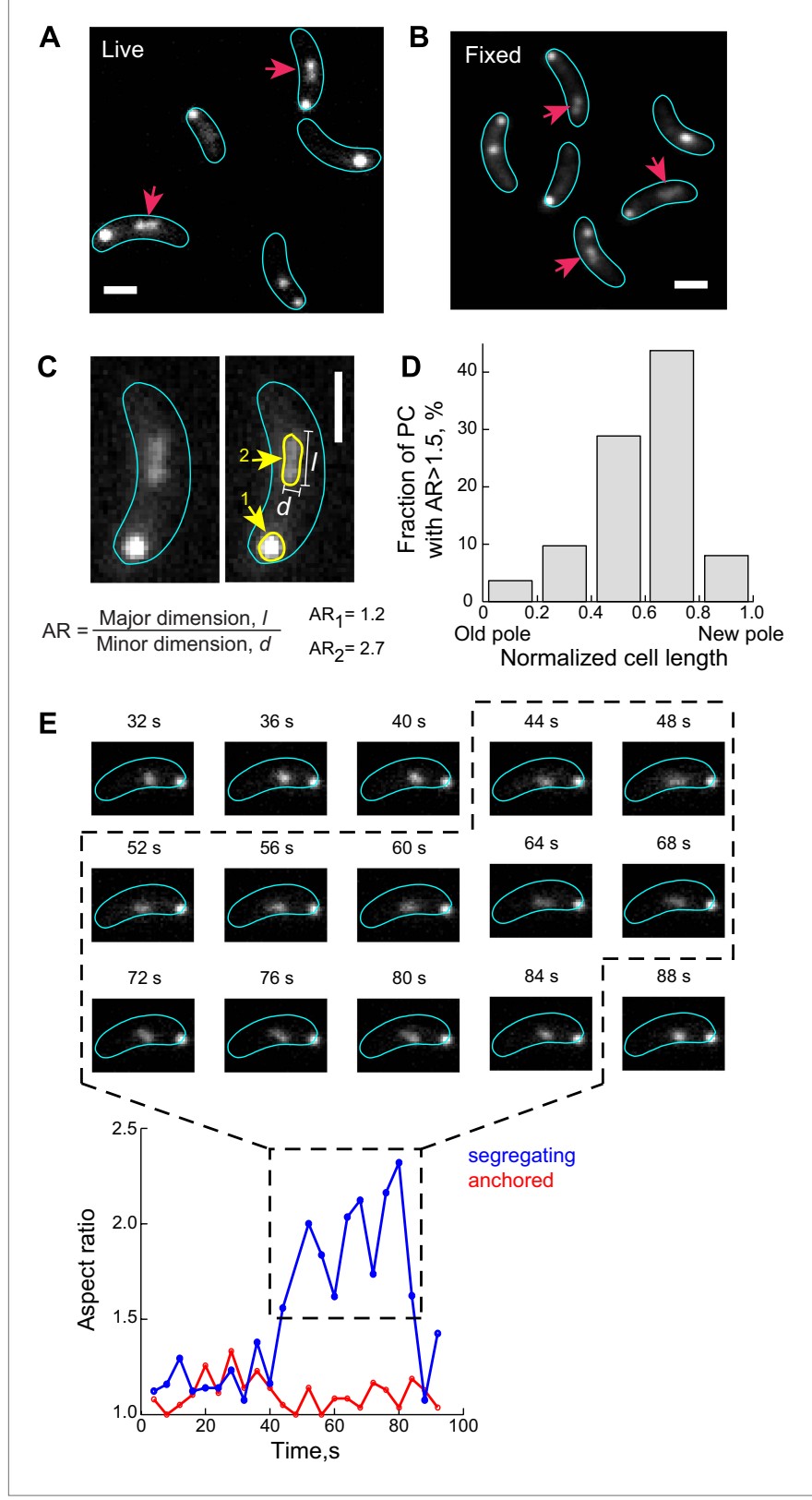

**Figure 6**. The ParB/*parS* complex transiently adopts an extended conformation during the fast segregation phase. (**A**) A fraction of GFP-ParB/*parS* complexes (red arrows) adopts extended conformation that appears as non-diffraction limited spots in fluorescent images of live CJW4762 cells. (**B**) Same as (**A**) except that cells were fixed with 4%
*Figure 6. Continued on next page*

*Figure 6. Continued*

formaldehyde prior imaging. (**C**) The aspect ratio (AR) of CFP-ParB/*parS* complexes in live CJW3367 cells was calculated as the ratio of the longest dimension, *l*, to the shortest dimension, *d*, for each partition complex signal. A representative cell is shown with the fluorescent CFP-ParB/*parS* signals outlined in yellow. (**D**) The propensity of a CFP-ParB/*parS* partition complex (PC) to display a decompacted conformation is shown with respect to cellular position. Individual CFP-ParB/*parS* complexes were binned according to their relative positions in the cell with old pole = 0 and new pole = 1 (see 'Materials and methods' for pole discrimination). The fraction of CFP-ParB/*parS* complexes with AR >1.5 in each bin are shown. (**E**) *Top*, representative time-lapse sequence (4-s intervals) showing de-compaction and recoiling of a segregating partition complex (marked by GFP-ParB) in a CJW4762 cell. *Bottom*, ARs of the anchored (red) and segregating (blue) ParB/*parS* complexes as a function of time is shown. For all the experiments indicated above, the expression of ParB fusion proteins was induced with 0.03% xylose in M2G for 60–75 min prior to synchronization and imaging. All scale bars = 1 μm.

## The proper range of ATPase activity is important for efficient and robust translocation

The DNA-relay model suggests that the ParABS system utilizes intrinsic DNA dynamics inside cells to translocate a cargo. It also suggests that the role of a ParA-ATP dimer is to provide a timed and reversible connection between the partition complex and the dynamic DNA. According to the model, a fluctuating DNA locus is captured out-of-equilibrium by the partition complex (via ParA-ATP dimers). As the DNA locus returns to its equilibrium, it brings the partition complex along. The association time between ParB/*parS* and the DNA-bound ParA-ATP dimer—which is defined by ParB-stimulated ATP hydrolysis rate ($k_{cat}$)—is critical for this mechanism to work efficiently. Under optimal conditions, the partition complex would be released immediately once equilibrium is reached to catch the next out-of-equilibrium DNA-bound ParA-ATP dimer. This would result in an apparent relay of the partition complex from one DNA-bound ParA-ATP dimer to another. A back-of-the-envelope calculation ('Materials and methods') estimates that the optimal attachment time would be 36 s. Thus, the optimal rate of ATP hydrolysis, which is the inverse of this optimal attachment time, should be about 0.03 s$^{-1}$. This value is in remarkably good agreement with the ParB-induced ATP hydrolysis rate ($k_{cat} = 0.03$ s$^{-1}$) we obtained from our biochemical data (*Figure 1F*).

We expect that if the ATP hydrolysis were too slow, the partition complex would remain bound to a ParA-ATP dimer too long, slowing down progression. Conversely, too high ATP hydrolysis rate would result in premature release of the complex (before it reached the equilibrium point), under-utilizing the potential of the elastic force. To examine how the performance of the DNA-relay model depends on $k_{cat}$, we varied the $k_{cat}$ values in our simulations. We found that for the estimated $D_{PC} = 0.0001$ μm$^2$ s$^{-1}$ (*Figure 4—figure supplement 2*), decreasing or increasing $k_{cat}$ by 10-fold negatively impacts the speed and robustness of translocation (*Figure 7A,B*). This suggests that the hydrolysis rate has been optimized during evolution for the proposed DNA-relay mechanism.

## Discussion

Central to the mechanism of ParABS-dependent transport is the origin of the translocation force. We did not find any evidence supporting a eukaryotic-like filament-based mechanism. Rather, our experiments and simulations collectively suggest a DNA-relay mechanism, in which the DNA-associated ParA-ATP dimers serve as transient tethers that harness the intrinsic dynamics of the chromosome to relay the partition complex from one DNA region to another. In this model, the translocation force is derived from the elastic property of the chromosome. Elastic dynamics may be inherent to highly compacted and ordered DNA as they have also been observed for chromosomal loci in eukaryotic nuclei (*Verdaasdonk et al., 2013*) as well as in nucleoids isolated from lysed *E. coli* cells (*Cunha et al., 2005*). Furthermore, a modeling study has shown that the *E. coli* chromosome behaves like a spring filament based on the positional distribution of chromosomal loci across cells (*Wiggins et al., 2010*).

A recent theoretical study suggests that a ParA gradient with ParB binding alone may generate a thermodynamic force (called chemophoresis) (*Sugawara and Kaneko, 2011*). While such force may contribute, it is not clear whether the in vivo conditions (e.g., ParA concentration, cargo size, time scale) can result in a thermodynamic force large enough to account for the translocation kinetics observed inside cells. On the other hand, we show here that the DNA-relay mechanism is sufficient to produce the sub-piconewton force that we estimated for ParA-dependent translocation.

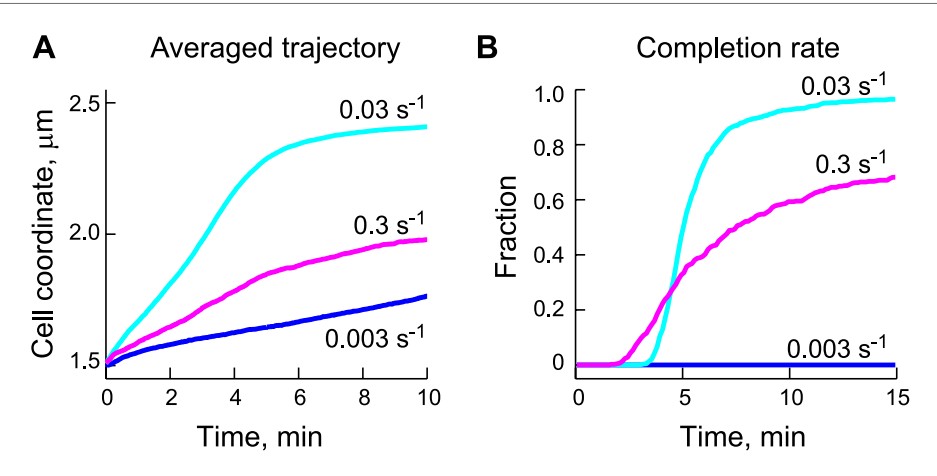

**Figure 7**. Appropriate ParB-stimulated ParA ATPase rates are important for the robustness of the DNA-relay model. (**A**) Averaged trajectories of the partition complex along the long cell axis during the fast ParA-dependent phase were simulated using varied ParB-stimulated ParA ATPase rates ($k_{cat}$) and a fixed diffusion coefficient for the ParB/parS complex ($D_{PC}$) of 0.0001 μm²/s. (**B**) Same data set as (**A**), except that for each $k_{cat}$, the fraction of trajectories that completed translocation are shown as a function of time.

The dynamic nature of chromosomes seem to be a universal feature of active cells (***Marshall et al., 1997***; ***Weber et al., 2010***, ***2012***; ***Javer et al., 2013***; ***Verdaasdonk et al., 2013***; ***Zidovska et al., 2013***). Since many ParA-like proteins involved in positioning of various large cytoplasmic cargos (chromosomal regions, plasmids, chemotaxis clusters) have a conserved and functionally important DNA-binding surface (***Hester and Lutkenhaus, 2007***; ***Castaing et al., 2008***; ***Ptacin et al., 2010***; ***Schofield et al., 2010***; ***Roberts et al., 2012***), the DNA dynamics may also be exploited in these systems to achieve active partitioning.

Another common property of all ParABS systems examined biochemically so far is the relatively low ParA ATPase activity, even after ParB stimulation. Our data suggest that ATPase activity is near-optimal for cargo translocation via the DNA-relay mechanism (***Figure 7***).

ParA-like proteins have been implicated in other biological functions besides partitioning (***Thanbichler and Shapiro, 2006***; ***Murray and Errington, 2008***). For example, *B. subtilis* Soj controls DNA replication initiation in vegetative cells (***Murray and Errington, 2008***). While this distinct function involves interactions with other cellular components (e.g., DnaA), biochemical evidence suggests that differential regulation of ParA dimeric states by ParB also plays an important role. As shown here, *C. crescentus* ParA ATPase activity requires high concentration of ParB (***Figure 1F***), which insulates ParA-ATP dimers from premature turnover by diffusing ParB and restricts stimulation of ATP hydrolysis to the ParB-rich partition complex. In contrast, in *B. subtilis*, a low concentration of Spo0J (ParB) is sufficient to stimulate Soj ATPase activity, such that even diffusing Spo0J promotes ATP hydrolysis (***Scholefield et al., 2011***). As a consequence, in *C. crescentus*, ParA accumulates as a cloud of DNA-bound ParA-ATP dimers, primed for the transport of the ParB/parS complex, whereas in *B. subtilis*, Soj mostly exists in a replication-inhibitory monomeric form that cannot bind the DNA (***Scholefield et al., 2011***). Consistent with this notion, a mere overproduction of Soj, which should tip the balance toward the Soj dimer state, results in cloud-like localization of Soj and Spo0J/parS translocation in the receding wave of the Soj cloud (***Marston and Errington, 1999***; ***Quisel et al., 1999***). Thus, a common biochemical framework can be tuned to generate different system behaviors.

In conclusion, we present a new physical mechanism for intracellular transport in which the chromosome plays a mechanical function. We anticipate that other facilitating or prohibiting elements (e.g., replication, transcription, and entropy-driven polymer separation, chromosome topology and glassy cytoplasm) (***Jun and Mulder, 2006***; ***Le et al., 2013***; ***Wang et al., 2013***; ***Parry et al., 2014***), which were not considered in our DNA-relay model, have an influence on chromosome partitioning. Further quantitative knowledge of these effects will allow us to build upon our DNA-relay model to gain a more comprehensive understanding of chromosome segregation in bacteria.

## Materials and methods

### Strains, protein fusion functionality and culture conditions

Strains used in this study and methods of strains construction are detailed in Table S1 in *Supplementary file 1*. Oligonucleotide primers used in this study are tabulated in Table S2, which is presented in *Supplementary file 2* together with methods of plasmids construction. Synchrony, conjugation, transformation and transduction with the bacteriophage ΦCR30 were performed as previously described (*Ely, 1991*). *C. crescentus* strains were grown at 30°C in the defined minimal M2G medium (0.87 g/l $Na_2HPO_4$, 0.54 g/l $KH_2PO_4$, 0.50 g/l $NH_4Cl$, 0.2% [wt/vol] glucose, 0.5 mM $MgSO_4$, 0.5 mM $CaCl_2$, 0.01 mM $FeSO_4$) unless otherwise stated. Since none of the *C. crescentus* strains used in this study contained replicative plasmids, antibiotics were omitted when cells were grown for the purpose of imaging. For all experiments, cells were harvested from exponentially growing cultures.

ParA and ParB are essential for viability in *C. crescentus* (*Mohl and Gober, 1997*). The *parA-dendra2* (this study), *parA-eyfp* (*Schofield et al., 2010*) and e*gfp-parB* (*Thanbichler and Shapiro, 2006*) fusions used in this study support viability when expressed as the only copy, indicating that they are functional. We previously showed that *parA-eyfp* is functional (*Schofield et al., 2010*). In the M2G medium at 30°C, the doubling times for the strains expressing *parA-dendra2* or e*gfp-parB* are 133 ± 2 min and 142 ± 2 min, respectively, as compared to 133 ± 2 min for wild-type CB15N grown under the same condition. For the strains in which the synthesis of ParA and ParB protein fusions was induced from the xylose-inducible promoter (pXyl) as a second copy, cells were grown in the presence of 0.03% xylose for 60–75 min (fluorescent ParB fusions) or 0.3% xylose for 1 hr (ParA-Dendra2). Under these inducing conditions, the localization of these second-copy fusions was visually undistinguishable from that of single copies (data not shown).

### Purification of ParA derivatives

For the purification of ParA or $ParA_{R195E}$, BL21(DE3) cells carrying pET24HT-ParA or pET24HT-ParA(R195E) were grown in Luria Broth supplemented with 50 µg/ml kanamycin to an $OD_{600}$ = 0.6, chilled to 18°C before IPTG (1 mM) was added to induce His6-ParA synthesis overnight at 18°C. Cells were collected by centrifugation at 5000×*g* for 20 min and stored at −80°C. The cell pellet was resuspended in buffer A1 (100 mM Hepes/KOH, 100 mM KCl, 1 mM EDTA and 10% glycerol) supplemented with EDTA-free Roche protease inhibitor, 1 mM DTT, 0.5 mM Mg-ATP, 1 kU DNase I and 0.5 mg/ml lysozyme. The cell suspension was incubated on ice for 30 min before 4 M KCl was added to a final concentration of 1 M (to dissociate ParA from the DNA). Following sonication, the sample was clarified by centrifugation at 100,000×*g* for 30 min. One molar imidazole (pH 7) was added to the clarified lysate to a final concentration of 40 µM. The supernatant was then passed through a Ni-NTA column (Qiagen, Germany) thrice by gravity flow. The resin was washed with 30 column volumes of buffer A2 (25 mM Hepes/KOH pH 7.4, 450 mM KCl, 50 mM potassium glutamate [KGlu], 1 mM $MgSO4$, 40 mM imidazole, 1 mM DTT and 100 µM MgATP) before elution was carried out with 50 ml of buffer A3 (25 mM Hepes/KOH pH 7.4, 450 mM KCl, 50 mM KGlu, 1 mM $MgSO4$, 300 mM imidazole, 1 mM DTT and 100 µM MgATP). Eluent was collected in 1-ml fractions. Fractions containing high concentrations of protein (determined by Bradford assay using Bovine IgG as standards) were pooled. The pooled sample usually had ~2–2.5 mg/ml protein content as determined by Bradford assay. TEV protease was diluted into the sample at ~1:30 (wt/wt) ratio and incubated at room temperature for 2 hr to remove the 6 × His tag. The cleavage efficiency was approximately 90–95%. The sample was then concentrated approximately twofold using Amicon Ultra-14 (MWCO = 10 kDa) and subjected to gel filtration fractionation on a Superdex 75 16/60 column (GE Healthcare Life Sciences, Pittsburgh, PA) at 1 ml/min flow rate in buffer A4 (25 mM HEPES/KOH pH 7.5, 200 mM KGlu, 1 mM $MgSO_4$, 1 mM DTT, 100 µM Mg-ATP and 5% glycerol). Residual His-tagged proteins were removed by passing the sample through a HisTrap HP column (GE Healthcare Life Sciences) in buffer A4 supplemented with 40 mM imidazole. A pooled sample was concentrated to ~1.5 mg/ml and subjected to dialysis against buffer A4 supplemented with 20% glycerol overnight at 4°C, and stored in small aliquots at −80°C.

Polyhistidine-tagged $ParA_{G16V}$-YFP was purified using $Ni^{2+}$ column as described for wild-type protein with the following modifications. To remove the imidazole, we induced $ParA_{G16V}$-YFP-His6 precipitation by dialyzing the sample into buffer A3 and re-solubilized the precipitated protein (which we isolated by centrifugation) in buffer A1 supplemented with 1 M KCl and 1 mM DTT.

## Purification of ParB variants

BL21(DE3)/pET21b-ParB cells or BL21(DE3)/pET21b-ParB(L12A) cells grown up to $OD_{600}$ = 0.6 were incubated with 1 mM IPTG for 4 hr at 37°C. Cells were collected by centrifugation at 5000×$g$ for 10 min at 4°C and frozen at −80°C. Cells were resuspended in buffer B1 (50 mM Hepes/KOH pH 7.0, 25 NaCl, 0.1 mM mM EDTA, 5 mM $MgCl_2$ and 1 mM DTT) supplemented with 1 kU DNase I and Roche EDTA-free protease inhibitor. Cell lysis was induced by passing the sample through a French press twice at 16,000 psi. Following 30 min incubation on ice, the sample was clarified by two rounds of centrifugation at 50,000×$g$ for 25 min. The clarified lysate was injected into an HiPrep SP 16/10 column (GE Healthcare Life Sciences) equilibrated in buffer B1 at 2 ml/ml and washed with four column volumes (CV) of buffer B1 before a gradient (50–500 mM NaCl) was developed over 20 CV with buffer B2 (50 mM Hepes/KOH pH 7.0, 1 M NaCl, 0.1 mM EDTA, 5 mM $MgCl_2$ and 1 mM DTT). Fractions containing ParB-His6 were pooled and imidazole was added to ~15 mM before incubating the sample with NiNTA resin (Qiagen) for 1 hr. The mixture was poured into the column to remove unbound proteins. The resin was washed with more than 20 CV of buffer B3 (25 mM Hepes/KOH pH 7.0, 300 mM NaCl, 5 mM $MgCl_2$, 40 mM imidazole and 1 mM DTT) followed by 2 CV of buffer B3 minus imidazole, and eluted with buffer B4 (25 mM Hepes/KOH pH 7.0, 300 mM NaCl, 5 mM $MgCl_2$, 300 mM imidazole and 1 mM DTT). Fractions containing ParB-His6 were pooled and subjected to dialysis in 100 × sample volume of buffer B5 (25 Tris/HCl pH 7.5, 50 mM NaCl, 5 mM $MgSO_4$, 0.1 mM EDTA and 1 mM DTT) overnight at 4°C. The sample was clarified at 30,000 × $g$ for 15 min prior to injection into an HiPrep Heparin 16/10 column equilibrated in buffer B5 at 1.5 ml/min. The column was washed with 5 CV of buffer B5 before a gradient (50–250 mM NaCl) was developed over 20 CV with buffer B6 (25 Tris/HCl pH 7.5, 1000 mM NaCl, 5 mM $MgSO_4$, 0.1 mM EDTA and 1 mM DTT). Fractions containing ParB-His6 were concentrated in a Amicon-15 unit and chromatographed over a Superdex 75 16/60 column equilibrated in buffer B7 (25 mM Hepes/KOH pH 7.5, 150 mM KGlu, 5 mM $MgSO_4$, 0.1 mM EDTA and 1 mM DTT) at a flow rate of 0.75 ml per min. Pooled samples were subjected to dialysis with two changes of 1 l of buffer B7 + 20% glycerol and stored at −80°C in small aliquots.

## Gel filtration analysis

ParA samples in buffer A4 + 20% glycerol were incubated with 25 mM EDTA for 1 hr at 37°C before their buffer was exchanged with buffer C (25 mM Hepes/KOH pH 7.5, 150 mM KGlu, 2.5 mM EDTA and 1 mM DTT) using a spin column to remove ATP. The samples were mixed with 2.5 mM Mg-ATP (for '+ATP' reactions) or buffer (for '−ATP' reactions), incubated at room temperature for 10 min, and centrifuged at 16,000×$g$ for 30 s. 50-microliter samples of ParA at approximately 50 μM were injected into a Superdex 75 10/300 GL column (GE Healthcare Life Sciences) equilibrated with 25 mM Hepes/ KOH pH 7.5, 150 mM KGlu, 5 mM $MgSO_4$, 2 mM Mg-ATP, 0.1 mM EDTA and 1 mM DTT for the +ATP reactions or 25 mM Hepes/KOH pH 7.5, 150 mM KGlu, 2.5 mM EDTA and 1 mM DTT for the −ATP reactions at a flow rate of 0.6 ml/min. The elution profiles were monitored at an absorbance of 280 nm.

## SEC-UV/LS/RI analysis

To examine whether ParB forms dimers in solution, we performed a size exclusion chromatography (SEC) coupled with UV, on-line laser light scattering (LS) and refractive index (RI) detectors (SEC-UV/ LS/RI). Specifically, purified His6-ParB (pre-filtered through a 0.22 μm filter) was applied on a Superose 6, 10/30, HR SEC column (GE Healthcare Life Sciences) equilibrated in 100 mM HEPES, pH 7.4, 150 mM NaCl, 1 mM DTT buffer and chromatographed at a flow rate of 0.3 ml/min. Elution from SEC was monitored by a photodiode array (PDA) UV/VIS detector (996 PDA, Waters Corp., Milford, MA), differential refractometer (OPTI-Lab, or OPTI-rEx Wyatt Corp., Santa Barbara, CA), and static, multi-angle laser light scattering detector (DAWN-EOS, Wyatt Corp.). The Millennium software (Waters Corp.) controlled the HPLC *Alliance* 2965 (Waters Corp.) system, to which the column was connected to, and data collection from the multi-wavelength UV/VIS detector, while the ASTRA software (Wyatt Corp.) collected data from the refractive index detector, the light scattering detectors, and recorded the UV trace at 280 nm sent from the PDA detector. Data collection and analyses were carried out at the Keck Foundation Biotechnology Resource Laboratory, Yale University.

## ATPase assay

The ATPase activity of ParA and ParA$_{R195E}$ was measured using an ATP/NADH-linked assay (*De La Cruz et al., 2000*), which we modified for a 96-well plate format. Reactions (100 μl) containing the appropriate

concentrations of ParA, ParB and salmon sperm DNA (Invitrogen), 5 × NADH enzyme mix (310 µM NADH, 100 U/ml of lactic dehydrogenase, 500 U/ml pyruvate kinase, and 2.5 mM phosphoenolpyruvate), 25 mM Hepes/KOH (pH 7.4), 10 mM MgSO$_4$, 150 mM KGlu, 5% glycerol and 1 mM DTT were mixed with 2.5 mM ATP unless mentioned otherwise. Absorbance measurements at 340 nm were taken in Corning UV Transparent Flat Bottom 96-well plates in a Synergy 2 plate reader (BioTek, Winooski, VT) at 30°C at 1-min intervals. Initial velocities were calculated from a linear regression of each time course and corrected for spontaneous ATP hydrolysis and NADH oxidation. We also ran parallel control experiments to account for residual ATPase in ParB protein preparations. A standard curve with known amounts of NADH was obtained and used to convert the rate of ADP production from absorbance/time to concentration/rate.

## Determination of ParA and ParB concentrations in *C. crescentus* by quantitative Western blotting

We first estimated the number of ParB and ParA molecules by quantitative Western blotting of cell lysates. For ParB, swarmer cells of *C. crescentus* CB15N were isolated by differential centrifugation (*Evinger and Agabian, 1977*) and resuspended in M2G medium to an OD$_{660}$ of 0.5. One milliliter of cell culture was pelleted and used to make a cell lysate for Western blotting. The cell lysate was separated on an Any-kD Mini-PROTEAN TGX Precast Gel (Biorad, Hercules, CA), proteins were transferred to a nitrocellulose membrane that was probed with 1:50,000 dilution of anti-ParB polyclonal antibody (Proteintech Group, Chicago, IL) and 1:10,000 dilution of anti-rabbit secondary antibody (Biorad). Signals were developed with an enhanced chemiluminescence reagent (GE Healthcare Life Sciences) and detected by exposing the membrane to a Kodak film (Carestream Health, Rochester, NY) or using a Typhoon PhosphoImager (GE Healthcare Life Sciences). The intensities of bands were quantified using ImageJ and compared against a standard curve generated from known amounts of ParB-His6 probed on the same blots. Since, at low concentrations, we lost significant amount of ParB-His6 presumably due to nonspecific binding of the protein to the wall of the plastic tubes, the ParB-His6 standard stock was prepared by diluting purified ParB-His6 in SDS buffer denatured cell lysates made from strain MT174 (which produces GFP-ParB instead of native ParB). After normalizing for the fraction of lysate loaded, we calculated the number of ParB molecules in the CB15N lysate by dividing the ParB amount by the molecular weight of ParB and by multiplying by the Avogadro number. In parallel, the number of cells used to prepare the cell lysate used for the Western blots was determined by serial dilution followed by quantification of colony forming unit (CFU) on an agar plate containing PYE medium. Using lysates made from four different pellets, we determined that swarmer cells contain 720 ± 80 (mean ± SD) ParB molecules (or 360 ParB dimers) per cell.

The number of ParA molecules per cell was determined using the same protocol as described for ParB with the following modifications. In the absence of high quality anti-ParA antibody, we used JL-8 anti-GFP antibody (Clontech, Mountain View, CA) to quantify the amount of ParA-YFP in a *C. crescentus* strain (CJW3010) in which *parA* has been cleanly replaced by a *parA-yfp* fusion at the native chromosomal location (*Schofield et al., 2010*). Varying amount of purified ParA-YFP-His6 (added to cell lysate prepared from wild-type cells) was used to create a calibration curve. We used 1,000 × dilution of JL-8 as primary antibody and 10,000 × dilution of goat anti-mouse secondary antibody (Biorad). Since the dynamics of ParB/*parS* segregation in the ParA-YFP-producing CJW3010 strain is indistinguishable from that of the wild-type, we assume that wild-type cells contain similar concentration of ParA molecules.

To convert the number of molecules per cell to concentration, we divided the number of molecules per cell by the Avogadro number and the estimated volume of a typical swarmer cell. The cytoplasmic volume of a swarmer cell, 0.25 fL, was obtained by cryo-tomography from 3D segmentation of a *C. crescentus* swarmer cell (*Briegel et al., 2006*). The concentrations of ParA and ParB were calculated using the formula = # Molecules/Avogadro Number × Volume M.

## Wide-field fluorescence imaging and image processing

For all microscopy observations, cells were spotted on 1% agarose pads containing M2G medium, unless specified otherwise. Images were acquired using an Eclipse Ti-U microscope (Nikon, Tokyo, Japan) with an Orca-ER camera (Hamamatsu Photonics, Hamamatsu City, Japan) and phase-contrast objective Plan Apochromat 100 × /1.40 NA (Carl Zeiss, Oberkochen, Germany) at room temperature except for time-lapse experiments (30°C). Images were acquired and processed with either

MetaMorph software (Molecular Devices, Sunnyvale, CA) or MATLAB (The MathWorks, Natick, MA). Cell mesh creation and fluorescence quantification were done using the open source, MATLAB-based software MicrobeTracker (*Sliusarenko et al., 2011*). Identification of diffraction-limited fluorescence spots was done using SpotFinder, an accessory in MicrobeTracker, unless mentioned otherwise.

## Quantification of ParB subcellular distribution

To measure the fraction of ParB molecules associated with the chromosomal *parS* region, we used a strain (MT174) in which the *parB* gene has been substituted by a functional *parB-gfp* fusion at the native chromosomal location without affecting the operon structure (*Thanbichler and Shapiro, 2006*). MT174 cells and wild-type CB15N cells, both from exponential phase cultures, were spotted on the same pad and imaged. Cells were segmented using MicrobeTracker (*Sliusarenko et al., 2011*). The signal intensity (in the GFP channel) of each cell normalized by its area was plotted as a histogram, which clearly showed two populations of cells (*Figure 1—figure supplement 3B*). A bimodal Gaussian model was used to fit the data. Autofluorescence in MT174 cells producing GFP-ParB was subtracted by the normalized signals detected in CB15N cells (lacking a GFP fusion). We used MicrobeTracker to quantify the fluorescence associated with segmented areas along the long cell axis. We assumed that segments with the least fluorescence concentration (fluorescence within the segment divided by the segment area) correspond to cell area far away from the GFP-ParB/*parS* complex and therefore their fluorescence values report the concentration of freely diffusing GFP-ParB. For each cell, we computationally ranked the fluorescence concentration within each cell segment, selected the lowest 10% of the total segments containing the lowest fluorescence concentration, calculated the average, and normalized this value to the total cell area to obtain the fluorescence value associated with the total diffusing GFP-ParB. This value was divided by the total GFP-ParB signal for that particular cell to obtain the fraction of diffusing GFP-ParB signal. The fraction of GFP-ParB in the partition complex = (1−fraction of diffusing ParB) × 100%.

## Analysis of the trajectory of the partitioning complex

To induce the expression of *gfp-parB*, 0.03% xylose was added to an exponential culture of CJW4762 cells for 60–75 min prior to subjecting the culture to synchrony. Isolated swarmer cells were spotted on a 1.5% agarose pad and imaged at 30-s intervals. Cell outlines were acquired using MicrobeTracker and the positions of the partition complex spots within each cell were detected and computed using SpotFinder (*Sliusarenko et al., 2011*). The trajectory of each partition complex was built using a custom-written MATLAB script that minimizes the total displacements of the two spots between the current frame and the previous frame (*Supplementary file 3*). To analyze segregation, we considered only trajectories that reached the new pole. To identify the 'fast' translocation phase in each trajectory, we wrote a MATLAB script that first linearly interpolates and smoothes each trajectory along the long cell axis before calculating the second derivatives of the curve (*Supplementary file 4*). The two inflection points covering the longest distance travelled were defined as the start and end of the 'run' phase. All traces were inspected by eye and only traces with correctly assigned start and end points were kept (141 out of 155).

## Aspect ratio analysis of the fluorescent signal associated with the partition complex

Partition complexes were identified using a previously reported algorithm for detecting fluorescent objects (both diffraction-limited and non-diffraction limited) in images (*Parry et al., 2014*). The output was analyzed to compute the major axis length and minor axis length for each partition complex using the regionprops function in MATLAB. The aspect ratio (or AR, the ratio between the major axis length and the minor axis length) was calculated for each partition complex. Before plotting AR as a function of cell coordinate, we oriented each cell such that zero corresponded to the old pole using the localization of ParA-YFP as a reference.

## Cleaning glass slides and cover slips for single-molecule imaging

Glass slides and cover slips used for single-molecule fluorescence microscopy were washed in the following order: 15 min sonication in 1 M KOH, 15 min sonication in milliQ $H_2O$ and 15 min sonication in 70% ethanol with 3 × milliQ $H_2O$ rinses between solution changes. Cleaned glass slides and cover slips were then dried with pressured air and used within the same day.

## Fluorescence-based quantification of ParA-YFP abundance in live *C. crescentus* cells

### Measurement of single ParA-YFP fluorescence

Cleaned coverslips were treated with 0.1% poly-Lysine solution on one side for 30 min, washed extensively with buffer B7 supplemented with 10% glycerol and 0.5 mM DTT and dried with pressured air. To quantify the fluorescence intensity of single ParA-YFP molecules, we used a solution of $ParA_{G16V}$-YFP-His6 (500 pM). This reagent was used because of its availability at the time of the experiment and the G16V mutation on ParA should not affect the fluorescent property of YFP. One hundred microliters of the $ParA_{G16V}$-YFP-His6 solution was incubated on the poly-Lysine coated side of the coverlip for 10 s and washed extensively with buffer B7 supplemented with 10% glycerol and 0.5 mM DTT. Most of the wash solution was removed by contacting the side of the coverslip with a sheet of Kimwipe, leaving behind ~1–3 µl. The coverslip was then mounted onto a cleaned glass slide and sealed. The images of the surface-immobilized YFP molecules were stream-acquired with 2-s integration time. In these images, $ParA_{G16V}$-YFP-His6 molecules appeared as distinct diffraction-limited fluorescent spots. For image analysis, we centered a 9 × 9 pixel box on each diffraction limited spot and integrated the fluorescence intensity in the box over time (*Figure 1—figure supplement 6A,B*). To determine the fluorescence intensity of a single $ParA_{G16V}$-YFP-His6 (or ParA-YFP) molecule, we analyzed only spots with a fluorescence profile that showed single-step photobleaching (*Figure 1—figure supplement 6B*, i). We determined the fluorescence intensity of a single YFP by calculating the difference in averaged fluorescence intensities before and after the stepwise photobleaching, *ΔI*. The mean fluorescence intensity of a single ParA-YFP molecule was determined from the mean of a Gaussian fit to the distribution of *ΔI* (*Figure 1—figure supplement 6C*).

### Measurement of total ParA-YFP fluorescence per cell

Exponential cultures of cells expressing *parA-yfp* from the *parA* native locus (CJW3010 strain) and wild type cells (CB15N which does not express *yfp*) at similar $OD_{660}$ were mixed at a 1:1 ratio before synchrony was performed to isolate swarmer cells. Wild-type cells were added to measure the fluorescence contribution from cellular autofluorescence. Swarmer cells of the two strains were then imaged on the same agarose pad using the same optical settings as those used to image single ParA-YFP molecules. Outlines of cells were obtained using MicrobeTracker. Fluorescence images were background-subtracted to remove signals from the agarose pad before further analysis. To completely capture the YFP fluorescence within each cells, we expanded the cell outline by five pixels. Cells with overlapping outlines were eliminated to prevent cross-cell 'fluorescence contamination'. Intensities of all pixels within each cell outline were then integrated and designated as total YFP fluorescence per cell. Number of ParA-YFP molecule per cell was calculated as the ratio between the total ParA-YFP fluorescence per cell and the mean fluorescence of single ParA-YFP molecules. We distinguished *parA-yfp*-expressing cells from wild-type cells through fluorescence thresholding (*Figure 1—figure supplement 6D*). In a separate experiment, we confirmed that this classification method reliably separates the two strains with better than 99% accuracy (data not shown).

## Super-resolution PALM microscopy

### Sample preparation and image acquisition

Cells were spotted on a 2% agarose pad made with 0.22 µm-filtered M2G. Imaging was performed at room temperature. All images were acquired on either an N-STORM microscope (Nikon) or a custom-built setup assembled on a commercial Axio Observer D1 microscope stand (Carl Zeiss). The N-STORM microscope was equipped with a CFI Apo TIRF 100 × oil immersion objective (NA 1.49), lasers emitting at 405 nm and 561 nm (Agilent Technologies, Santa Clara, CA) and a built-in Perfect Focus system. Raw single molecule data were taken in a field of view of ~43 × 43 µm² (256 × 256 pixels) with an Andor iXon X3 DU 897 EM-CCD camera (Andor Technology, South Windsor, CT) at a frame rate of 28.5–29 frames per second (fps). Under this acquisition condition, diffusing molecules appeared blurred and were rejected from our analysis. The custom-built microscope set up (*Huang et al., 2013*) was equipped with a 100 × /1.46-NA oil-immersion objective (alpha Plan-Apochromat 100 × /1.46 oil, Carl Zeiss), lasers emitting at 405 nm (50 mW, CrystaLaser, Reno, Nevada) and 568 nm (Innova 300, ~400 mW, Coherent, Santa Barbara, CA). Fluorescence was recorded with an ORCA-Flash 4.0 sCMOS camera (Hamamatsu Photonics) at a frame rate of 100 fps and a field of view of 23 × 23 µm².

## Camera calibration

To allow the conversion of camera output counts from our images into units of effective photons that follow a Poisson distribution, we calibrated the offset and amplification gain of the EM-CCD camera using a Poisson calibration routine (*Huang et al., 2011*). Briefly, the mean offset was determined from 1000 raw images using the 256 × 256 pixels region at the center of the EMCCD chip in a dark environment with no incident photons. To determine the amplification gain, we extracted the slope of the variance vs the mean scatter plot calculated from each pixel over 200 camera frames while illuminated with a stationary intensity pattern. This pattern was generated by imaging a slide containing 10–20,100-nm TetraSpeck microspheres (Invitrogen, Carlsbad, CA) in an out-of-focus plane such that an intensity pattern emerged from the overlapping point-spread-functions of the beads (*Lidke et al., 2005*; *Smith et al., 2010*; *Huang et al., 2011*). The sCMOS camera was calibrated as previously described (*Huang et al., 2013*).

## Analysis and reconstruction

Raw images were first corrected with the calibrated offset and gain values to convert pixel intensities into effective photon counts. The corrected images were analyzed using a previously described maximum-likelihood estimation algorithm to localize well-defined single emitters (*Smith et al., 2010*). Log-likelihood ratio test was used to retain only fitted emitters that contain >100 photons, and >2.5% corresponding p-value (*Huang et al., 2011*). The average localization precision for all dataset is approximately 19 nm (±6 nm, SD). The super-resolution images were reconstructed by placing 2D Gaussians (σ = 10 nm) on images with 9 nm pixel size. Drift corrections were performed on localized single molecule centers using cross-correlation method by assigning 500 frames for each correlation segment (*Mlodzianoski et al., 2011*). Images taken with the sCMOS camera was analyzed with a previously published sCMOS-specific script which takes into consideration sCMOS-intrinsic-pixel-dependent readout noise and achieves optimal precision at the theoretical limit (*Huang et al., 2013*). Images shown in *Figure 3* were generated from data acquired with the N-STORM, while those presented in *Figure 3—figure supplement 1* were taken with the custom-built setup.

## Width distribution analysis

Due to the large pixel size of our images, we first expanded the bright field images from 256 × 256 pixels to 1024 × 1024 pixels using bicubic interpolation before subjecting them to MicrobeTracker analysis to obtain cell outline. We then linked each single molecule to each cell outline by using the inpolygon function in MATLAB and calculated the distance for each single molecule to the cell center line as determined from the MicrobeTracker outline. To determine the individual distribution of Dendra2 fusions inside cells along the cell width, we selected cells that contain high density of localized emitters (>1000 per cell) for analysis. To analyze the distribution of emitters along the short cell axis, a 0.5–0.8 µm long segment was manually selected for each cell. These segments corresponded to regions containing the ParA 'cloud' in cells expressing ParA-Dendra2 or areas where the crescentin-Dendra2 filament orientation was parallel to the long cell axis. For cells expressing L1-Dendra2, the segments were randomly selected since the distribution of emitters along the long cell axis was mostly homogenous. Only segments with at least 600 emitters were analyzed to minimize statistical error. For segments that passed the selection, single emitter positions with respect to the short cell axis were then normalized to the mean of the distribution in each cell and plotted as a histogram.

## Estimation of DNA concentration inside *C. crescentus*

$$\text{DNA mass per } C.\,crescentus \text{ chromosome} = \text{Total } bp \text{ per chromosome} \times \text{mass per } bp$$

$$= 4 \times 10^6 \, bp \times 650 \, Da \, \text{bp}^{-1}$$

$$= 2.6 \times 10^9 \, Da \times \frac{1g}{6.022 \times 10^{23} \, Da}$$

$$= 4.31 \times 10^{-15} \, g$$

$$\text{Thus, in vivo DNA concentration} = \frac{\text{DNA mass}}{\text{Cell volume}} = \frac{4.3 \times 10^{-15} \, g}{2.5 \times 10^{-13} \, \text{ml}} = 17 \, \text{mg ml}^{-1}$$

### Estimation of the diffusion coefficient of the partition complex, $D_{PC}$

To estimate $D_{PC}$, we analyzed the movement of the ParB/*parS* complex in the slow, ParA-independent phase (*Shebelut et al., 2010*), which appears to exhibit diffusive behaviors. As shown in *Figure 4—figure supplement 2A*, we identified three 'segments' within the slow phase for each cell:

'Segment 1' corresponds to the part of the trajectory when the ParB/*parS* complex has detached from the old pole but has not yet duplicated. To identify this segment, we computationally looked for the part of trajectory in which there was only one ParB/*parS* complex in the cell and its position was at least 320 nm from the old cell pole.

'Segment 2' corresponds to the part of the trajectory when the ParB/*parS* complex proximal to the old pole has separated from its sister but has not yet re-attached to the old pole. This segment started 1 min after two ParB/*parS* complexes became visibly separated (to exclude possible active splitting mechanism) and ended when the complex moved within 320 nm from the old pole.

'Segment 3' corresponds to the part of the trajectory when the ParB/*parS* complex distal to the old pole has separated from its sister and has not engaged in the directed 'fast' translocation phase. This segment contained the maximum unbiased stretch of the trajectory 1 min after two ParB/*parS* complexes became visibly separated. We identified this segment by determining the longest part of the trajectory that remains confined within a virtual 500-nm wide box (i.e., no net directional motion).

We combined data from all trajectories for each of these three segments separately and analyzed their respective displacement distributions and mean square displacements (MSDs) (*Figure 4—figure supplement 2B,C*). We estimated $D_{PC}$ using two different approaches. First, we performed a Gaussian fitting (which is expected for normal diffusion) to the displacement distributions, which returned best fit $D_{PC}$ values of 0.0007 ± 0.0001, 0.00012 ± 0.0001 and 0.00012 ± 0.0001 $\mu m^2 s^{-1}$ for ParB/*parS* motion during segments 1, 2 and 3, respectively (*Figure 4—figure supplement 2B*). Consistent with these estimations, a linear fit to the first four time points of each MSD gave $D_{PC}$ values of 0.0004 ± 0.0003, 0.0001 ± 0.0006 and 0.00008 ± 0.0001 $\mu m^2 s^{-1}$ for segments 1, 2 and 3, respectively (*Figure 4—figure supplement 2C*). These values are close to the diffusion coefficient ($1.5 \times 10^{-4}$ $\mu m^2 s^{-1}$) of *E. coli* chromosomal *ori* region that we estimated from data presented in *Kuwada et al. (2013)*. In addition, the good linear MSD fits suggest that the motions of the unattached ParB/*parS* complex can be approximated as a normal diffusion at a minute scale (i.e., on the time-scale of segregation). Therefore, unless specified otherwise, we used $D_{PC} = 1 \times 10^{-4}$ $\mu m^2 s^{-1}$ (the average value for 'Segment 2' and 'Segment 3' given by the two approaches) in all simulations.

### Estimation of the diffusion coefficient of DNA-bound ParA-ATP dimers, $D_A$

We estimated $D_A$ from published results for generalized (i.e., time-dependent) diffusion coefficient of various chromosomal loci in *E. coli*, $D_{app} \sim (0.25–1.5) \times 10^{-3}$ $\mu m^2 s^{-0.4}$ (*Weber et al., 2012*; *Javer et al., 2013*). This gives an effective diffusion coefficient at 1-ms time scale (time interval used in our simulations): $D_A = D_{app} \Delta t^{-0.6} \sim (1.5–9) \times 10^{-2}$ $\mu m^2 s^{-1}$. We used $D_A$ value of 0.01 $\mu m^2 s^{-1}$ in the simulations of our DNA-relay model. As a control, we simulated the dynamics of DNA-bound ParA dimers in the absence of other factors and found that this estimated $D_A$ value closely recapitulated the step-size and position distributions that are experimentally observed for chromosomal loci such as the LacI-labeled *groESL* locus (data not shown).

### Chromosomal locus tracking

CJW2966, CJW5466 and CJW5468 cells were grown at 30°C in M2G supplemented with 5 µg/ml kanamycin. Prior to imaging, synthesis of LacI-CFP was induced with 0.03% xylose (plus 20 µM IPTG to prevent adverse effects on DNA replication and cell division). After 60–80 min, cells were washed and spotted on 1.2% agarose containing M2G and 20 µM IPTG, and imaged at 2-s time intervals between frames. LacI-CFP-labeled chromosomal loci appeared as diffraction-limited spots. Their positions (relative to the long and short axes of each cell) were determined using the SpotFinder (*Sliusarenko et al., 2011*). Only trajectories with a minimal length of five frames were considered for quantitative analysis.

### Estimation of the elastic constant of the DNA locus

To exclude differences in equilibrium position due to cell-to-cell variability, we measured deviations ($\Delta x$) from the mean position of each individual trajectory. The probability $P(\Delta x)$ of a locus to fluctuate around its equilibrium point is given by:

$$P(\Delta x) \sim e^{-\frac{E(\Delta x)}{kT}}$$

$$(1)$$

where $k$ is the Boltzmann constant, $T$ is the absolute temperature, and $E(\Delta x)$ is the energy associated with the fluctuation. For an elastic force, $F = k_{sp} \Delta x$ and $E(\Delta x) = k_{sp}\Delta x^2/2$ (with $k_{sp}$ being the elastic constant) and the probability of observing a locus at a distance $\Delta x$ from the equilibrium is a Gaussian distribution (which is what we observe in experiments):

$$P(\Delta x) \sim e^{-\frac{k_{sp}\Delta x^2}{2kT}} \tag{2}$$

We fitted the experimental distributions separately for deviations along the short and long axes of the cell with one free parameter $\sigma$ (with $1/\sigma^2 = k_{sp}/kT$), obtaining $\sigma = 38 \pm 1$ nm and $64 \pm 2$ nm for the short and long axis, respectively. These values were used in the Brownian dynamics simulations.

## Brownian dynamics simulations

We considered three models in this study. The 'diffusion' model contains only the diffusion terms of the *Equation 3* below. The 'diffusion-binding' model also includes binding and un-binding of the ParB/*parS* complex to the DNA-bound ParA dimers, but no force terms (i.e., immobile DNA-bound ParA dimers). The 'DNA-relay' model contains all terms described in *Equation 3*, which takes into consideration diffusion, binding/unbinding, and elastic forces (i.e., dynamic DNA-bound ParA dimers). The parameter values used in each model are listed in the *Table 2* unless stated otherwise.

Two-dimensional Brownian dynamics (BD) simulations were performed using a second-order approximation (*Branka and Heyes, 1998*). At each time step, the coordinates of the ParB/*parS* complex and DNA-bound ParA-ATP dimers were calculated as follows:

$$x(t_i + \Delta t) = x(t_i) + R_0 + \frac{D}{kT}F\Delta t + \frac{D^2}{2(kT)^2}FF'^{\Delta t^2} + \frac{D}{kT}F'R_1 \tag{3}$$

The first three terms in *Equation 3* correspond to first-order approximations typically used in BD simulations (*Saxton, 2007*), where $\Delta t$ is the time step of simulations, $x(t_i)$ is the coordinate of ParB/*parS* or DNA-bound ParA dimers at the previous time point, $D$ is the diffusion coefficient for the component considered ($D_{PC}$ for the ParB/*parS* complex or $D_A$ for DNA-bound ParA dimers), $kT$ is the temperature in energy units, $R_0$ specifies the random displacement due to diffusion (i.e., random number from a Gaussian distribution with zero mean $<R_0> = 0$ and variance $<R_0^2> = 2D\Delta t$), $F$ is the force acting on the particle considered (partition complex or DNA-bound ParA-ATP dimer). The last two terms in *Equation 3* correspond to further expansion of the stochastic equations (*Branka and Heyes, 1998*), where $F'$ is the derivative of the force ($F' = dF/dx$) and $R_1$ is a random number specified by a Gaussian distribution (with zero mean $<R_1> = 0$ and variance $<R_1^2> = (2/3)D\Delta t^3$), which is correlated with $R_0$: $<R_0 R_1> = D\Delta t^2$. The force terms for DNA-bound ParA and ParB/*parS* are:

$$F_{ParA} = k_{sp}\left(x_{ParA} - x_{ParA}^0\right) \text{ and } F'_{ParA} = k_{sp} \tag{4}$$

$$F_{ParB} = \sum k_{sp}\left(x_{ParA} - x_{ParA}^0\right) \text{ and } F'_{ParB} = n_{AB}k_{sp} \tag{5}$$

respectively, where $k_{sp}$ is the spring constant, $x_{ParA}^0$ is the equilibrium point of ParA dimers bound to DNA, and $n_{AB}$ is the total number of ParA dimers bound to the partition complex. Note that *Equation 5* is the sum over all ParA dimers bound to the partition complex. Also, only the ratio $k_{sp}/kT$ was defined as a model parameter (determined from experiment, $1/\sigma^2 = k_{sp}/kT$) since all force terms depends only on this combination of $k_{sp}$ and $kT$.

In our models, the ParB/*parS* complex can be in one of two states: (1) freely diffusing (no force terms) or (2) in a complex with one or more DNA-bound ParA dimers (non-zero force terms). We modeled ParA dimers and ParB/*parS* complex as disks with $R_{ParA}$ and $R_{PC}$ equal to 2 and 50 nm, respectively. The partition complex binds to ParA every time their disks overlap. At any given time, ParA could be in one of three different states: DNA-associated, in a complex with ParB/*parS*, or freely diffusing. The motion of DNA-bound ParA was simulated via *Equations 3* and *4*. When a ParA dimer was associated with the partition complex, its coordinate increment was equal to the increment of the partition complex. The motion of freely diffusing ParA monomers was not explicitly simulated, since ParA monomers do not interact with the partition complex. The motion of freely diffusing ParA is relatively fast ($D_{A\text{-}free} >> D_A$) leading to a uniform random distribution across the cytoplasm within seconds. Free ParA

molecules bind to DNA with uniform probability with respect to the short cell axis. However, in the long-axis direction, we emulated ParA distribution observed in cells (*Figure 4—figure supplement 1*) by assuming that the probability of ParA to re-bind to DNA is a linearly increasing function between the current ParB/*parS* location ($x_{PC}$) and the new pole:

$$P_{ParA-DNA} = \frac{2}{(l - x_{PC})^2}(x - x_{PC}), \text{ for } x_{pc} < x < l,\qquad(6)$$

where $l$ is a cell length, and the coordinates are defined as $x = 0$ at the old ole and $x = l$ at the new pole.

The transition of ParA between different states was modeled as a stochastic process (exponential distribution of times) with average times $\tau_{db}$ (for ParA$_{free}$ → ParA$_{DNA\text{-}bound}$) and $\tau_{cat}$ (for ParA$_{ParB/parS\ bound}$ → ParA$_{free}$). The rate of ParA dissociation from the DNA upon interaction with the ParB/*parS* complex is equal to the maximum hydrolysis rate, $\tau_{cat} = 1/k_{cat} \approx 30$ s (*Figure 1F*), unless mentioned otherwise. Note that because of its slow rate, dissociation of ParA from the DNA due to the basal ATPase activity of DNA-bound ParA was neglected. Since for the plasmid P1 system, the rate of freely diffusing ParA monomers dimerizing and binding to DNA is thought to be comparable to ATP hydrolysis rate (*Vecchiarelli et al., 2010*), we assumed that $\tau_{db} = \tau_{cat} = 30$ s for all simulations shown here. Note that varying $\tau_{db}$ between 1-100 s did not significantly affect the results of our simulations (data not shown).

We defined the starting conditions using experimentally measured values. At the beginning of the simulation, the partition complex was placed at $x(0) = 0.8$ μm (measured from the old pole, *Figure 2B*), $y(0) = 0$. At the start of each simulation, ParA-ATP dimers were distributed randomly using *Equation 6* with $x_{PC} = x(0)$. Cell boundaries were modelled as a reflective 2.5 μm × 0.4 μm rectangular. All simulations were carried out using $\Delta t = 1$ ms for a total of 2000 s or until the partition complex crosses $x_{finish} = 2.6$ μm, which ever happened first.

## Calculation of the optimal ATP hydrolysis rate for the DNA-relay model

To estimate the optimal attachment time between the partition complex and ParA-ATP dimers, we calculated the average time ($\Delta t$) required for the partition complex to reach an equilibrium point since its association with a DNA-bound ParA dimer. A particle under elastic force ($F = k_{sp} \Delta x$ where $k_{sp}$ is the elastic constant) moves a distance $\Delta x$ in a viscous medium during an interval $\Delta t$:

$\Delta x = v \Delta t = \mu F \Delta t = (D/kT) k_{sp} \Delta x \Delta t$

Here $v$ is the velocity of the particle, $\mu$ is the particle mobility linked to the diffusion coefficient $D_{PC}$ as $\mu = D_{PC}/(kT)$, $k$ is the Boltzmann constant and $T$ is the absolute temperature. This allowed us to calculate the optimal time of ATP hydrolysis using $\sigma_{long}^2 = kT/k_{sp}$:

$\Delta t = kT/(D_{PC} k_{sp}) = \sigma_{long}^2/D_{PC} = (0.06 \text{ μm})^2/10^{-4} \text{ μm}^2 \text{ s}^{-1} = 36$ s.

## Acknowledgements

We thank Drs Enrique M De La Cruz, Wenxiang Cao and Michael J Bradley for help with setting up the NADH-linked ATPase assay, Dr Pingyong Xu for providing the template for *mEos3.2*, Drs Herald Herrmann-Lerdon and Nicole Foeger for providing the pET24dHT plasmid, Dr Patrick Viollier for strains carrying the 139_lac and 165_lac constructs, Jennifer R Heinritz and Dr Whitman Schofield for providing plasmid constructs, Drs Grant Jensen and Ariane Briegel for sharing unpublished calculations of the swarmer cell volume, Dr Gitte Ebersbach Charbon for providing the ParB dimerization data, staffs of the Yale University Faculty of Arts and Sciences High Performance Computing Center for support, and members of the Jacobs-Wagner laboratory for critical reading of this manuscript and for fruitful discussions.

## Additional information

### Competing interests
JB: Discloses significant financial interest in Vutara, Inc (Salt Lake City, UT). The other authors declare that no competing interests exist.

## Funding

| Funder | Grant reference number | Author |
|---|---|---|
| National Institutes of Health | R01 GM065835 | Christine Jacobs-Wagner |
| Howard Hughes Medical Institute | | Christine Jacobs-Wagner |
| Wellcome Trust | 095927/A/11/Z | Jörg Bewersdorf |
| Yale University | Edward L. Tatum Fellowship | Hoong Chuin Lim |
| Child Study Center, Yale School of Medicine (Yale Child Study Center) | James Hudson Brown-Alexander Brown Coxe Postdoctoral Fellowship | Fang Huang |
| Howard Hughes Medical Institute (HHMI) | EXROP Scholarship | Bruno Gabriel Beltran |

The funders had no role in study design, data collection and interpretation, or the decision to submit the work for publication.

## Author contributions

HCL, IVS, Conception and design, Acquisition of data, Analysis and interpretation of data, Drafting or revising the article; BGB, FH, Conception and design, Acquisition of data, Analysis and interpretation of data; JB, Conception and design, Drafting or revising the article; CJ-W, Conception and design, Analysis and interpretation of data, Drafting or revising the article

## Additional files

### Supplementary files

• Supplementary file 1. Strains used in this study and the methods of strains construction.

• Supplementary file 2. Oligonucleotide primers used in this study and methods of plasmids construction.

• Supplementary file 3. Custom-built MATLAB function (buildTrace.m) for constructing trajectories of ParB/*parS* complexes in single cells.

• Supplementary file 4. Custom-built MATLAB function (findFastSegPhase.m) for identifying the 'fast' phase in the segregating trajectory.

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
