## [Decision Letter]

Thank you for sending your work entitled “Evidence for a DNA-relay mechanism in ParABS-mediated chromosome segregation” for consideration at *eLife.* Your article has been favorably evaluated by Senior Editor Richard Losick and 3 reviewers, one of whom is a member of our Board of Reviewing Editors, and one of whom, Kerry Bloom (peer reviewer #3), has agreed to reveal her identity.

The Reviewing editor and the other reviewers discussed their comments before we reached this decision, and the Reviewing editor has assembled the following comments to help you prepare a revised submission:

In short, we would like to receive a revised manuscript that addresses the major point raised by Reviewer # 1, to provide additional evidence for elastic dynamics in other regions of the Caulobacter chromosomes and to expand the scope of the modeling, as suggested by Reviewer #2. It will also be useful to provide additional explanations for the scenario raised by Reviewer #2 on how the separation of the *parS* sites at oriC on the chromosome might affect your model.

With these revisions, your work will make an important contribution to the fields of bacterial chromosome biology and bacterial cell division.

*Reviewer #1*:

This excellent manuscript by Lim et al provides a new mechanism for DNA segregation in the bacterium Caulobacter crescentus by the ParABS system. Perhaps the most important finding is that ParA does not form filaments but that it participates in chromosome segregation through a modulation of its ATPase activity stimulated by highly concentrated ParB at *ParS* sequences. Through a combination of beautiful biochemical and cell biological experiments (including super resolution experiments) they establish the central thesis of the paper.

Modeling work reveals inadequacy of a diffusion-binding mechanism. The authors arrive at the conclusion that some of aspect of DNA structure itself might contribute to polar segregation of chromosomes. In short elastic dynamics of chromosomal DNA in conjunction with ParA-ATP gradient drives chromosome segregation. This is an exciting new parameter that enters their final model. In summary, the experiments are well done and support the overall conclusions and this paper deserves to be published in *eLife*.

Since elastic dynamics of DNA is proposed to provide chromosome segregational forces and since this is a new dimension introduced to describe DNA segregation, I would like to see if 1-2 other loci (in addition to GroESL) undergo similar elastic dynamics, to further bolster the notion that the Caulobacter chromosome behaves like an elastic filament. I do not know how easy/difficult it would be to generate a strain by integration of LacI into other loci and perform such a study in Caulobacter.

Reviewer #2:

This is a beautifully written paper describing interesting and important new insights into the mechanism of ParAB mediated plasmid and chromosome segregation. The experiments are superbly documented with very thorough controls. Apart from quantitatively documenting many features of the biochemistry and cell biology of the Caulobacter ParAB system, a “DNA relay” model is presented, which can explain the directional movement of the origin of replication, without the need for ParA filaments, which have been an important feature of most previous models.

My only concern/criticism is that the new model is, in principle, rather similar to that recently proposed by the Mizuuchi lab, based on in vitro experiments with a plasmid ParABS system. The main difference between the two models would seem to be the proposed role of intrinsic fluctuating motion of DNA-bound ParA-ATP dimers. Intuitively, it seems to me that the modelling would work just as well by a Mizuuchi-like process and that tethering the ParA molecules to DNA provides a matrix on which the polar gradient of ParA can be fixed. It would be interesting to see this alternative scenario modelled. The new model also does not take account of the fact that each oriC region has multiple ParB molecules bound, and that there are several separated *parS* sites, each of which could collide with a DNA-bound ParA dimer, facilitating the walk up the ParA gradient towards the cell pole. Perhaps the occasional “stretching” of ParB foci represents simultaneous interactions made by different *parS* assemblies?

My feeling therefore is that this paper is important because it reinforces the notion that filament formation by ParA may not be crucial for its function and provides strong in vivo support for a Mizuuchi-like diffusion-ratchet model (which could equally be called a DNA relay). I am less convinced by the proposed role for an elastic force.

Reviewer #3:

This is an excellent manuscript that describes a novel mode of chromosome segregation in *C. crescentus*. The paper addresses several important unanswered questions in the field, and provides significant new insights. The data are clearly presented, thorough, and the model is well-reasoned.

Specific comments:

The mode of chromosome segregation in many bacteria is poorly understood. The ParABS system has been proposed to be analogous to the eukaryotic spindle-based mechanisms using a filament of ParA to pull the ParB/*parS* complex along. To better understand how the system works in vivo, these investigators begin with determining the DNA-dependent ATPase activity. They find a different behavior in the *C. crescentus* vs. *B. subtilis* components that has important biological consequences.

The investigators count the number of molecules, which will be a critical parameter in the model. The counting method is well-described.

The filament model of ParA function predicts a particular trajectory of the translocation. The measured trajectory (Figure 2) indicates a random, fluctuating path, at odds with the current models. Furthermore, the molecular counts indicate the in vivo concentration to be too low to accommodate filament formation.

The investigators proceed to examine diffusion rates and find that at least with simple diffusion models and the measured parameters, the biased diffusion model does not lead to directional motion.

The significant discovery comes in the analysis and incorporation of chromosome dynamics into the model. The ParA complex is influenced by the fluctuations of a dynamic DNA matrix. The model is clear and the investigators have done an excellent job integrating the biochemical cycle of the ParComplex with the fluctuations of DNA into a novel mechanism that provides highly significant insight into an important biological problem.

---

## [Author Response]

*In short, we would like to receive a revised manuscript that addresses the major point raised by Reviewer # 1, to provide additional evidence for elastic dynamics in other regions of the Caulobacter chromosomes and to expand the scope of the modeling, as suggested by Reviewer #2. It will also be useful to provide additional explanations for the scenario raised by Reviewer #2 on how the separation of the* parS *sites at oriC on the chromosome might affect your model*.

Thank you for giving us the opportunity to submit a revised manuscript. We have added experimental evidence for the elastic dynamics of two other chromosomal regions and expanded the scope of the modeling, as requested by Reviewer 1 and 2, respectively. Regarding the multiplicity of *parS* sites and their separation, please note that in the *C. crescentus* chromosome, there are only two *parS* sites and they are separated by only 42 nucleotides. Fully stretched, 42 nucleotides would be at most 14 nm, making the scenario raised by Reviewer 2 unlikely. Nevertheless, we have considered this scenario and shown that it does not alter our conclusions. Please see below for details.

Reviewer #1:

*Since elastic dynamics of DNA is proposed to provide chromosome segregational forces and since this is a new dimension introduced to describe DNA segregation, I would like to see if 1-2 other loci (in addition to GroESL) undergo similar elastic dynamics, to further bolster the notion that the Caulobacter chromosome behaves like an elastic filament. I do not know how easy / difficult it would be to generate a strain by integration of LacI into other loci and perform such a study in Caulobacter*.

We agree with the reviewer, and have imaged and analyzed the motion of two additional chromosomal loci (139_lac and 165_lac) (Viollier et al., PNAS, 2004). We found that they displayed elastic dynamics similar to the *groESL* locus. The results have been added to Figure 5 (panel C). The text and figure legend have been modified accordingly.

Reviewer #2:

*Intuitively, it seems to me that the modelling would work just as well by a Mizuuchi-like process and that tethering the ParA molecules to DNA provides a matrix on which the polar gradient of ParA can be fixed. It would be interesting to see this alternative scenario modelled*.

We apologize for not being more explicit in the original submission, but our ‘diffusion-binding’ model does test a ‘Mizuuchi-like’ process. In that model, a polar gradient of ParA-ATP dimers is fixed on a static DNA matrix exactly as proposed by the reviewer. As shown in Figure 4, this diffusion-binding model does not lead to fast or robust segregation. Note that we were, at first, surprised by the results because we had the same intuition as Reviewer 2. In hindsight, the poor performance of the diffusion-binding model should be expected because in this model, the movement of the partition complex is governed by diffusion. There is no force involved and the ParA-ParB interactions only stall the partition complex without productively moving the partition complex in the ‘right’ direction.

There is nothing in this mechanism that prevents the partition complex from diffusing in the wrong direction when the interaction between ParA and ParB is lost (i.e., in the ParA-depleted region away from the ParA gradient). This can be seen in Video 2. As a result, the diffusion-binding model does not perform better than the model in which the partition complex only passively diffuses (no interaction with ParA-ATP dimers). In fact, the diffusion-binding model quantitatively performs worse than the diffusion-alone model because the only thing the ParA/ParB interaction does is to temporally stops the motion of the partition complex, slowing down its ultimate arrival at the opposite cell pole. We found that it was essential to include the elastic dynamics of the underlying DNA (which had not been considered before) to explain the translocation kinetics observed in vivo. We have revised the manuscript to be more explicit.

*The new model also does not take account of the fact that each oriC region has multiple ParB molecules bound, and that there are several separated parS sites, each of which could collide with a DNA-bound ParA dimer*, *facilitating the walk up the ParA gradient towards the cell pole. Perhaps the occasional “stretching” of ParB foci represents simultaneous interactions made by different parS assemblies?*

Our modeling does take into account that multiple ParB dimers bind to the *parS* (ori) region and that the ParB-rich partition complex can interact with more than one DNA-bound ParA-ATP dimer. The interaction radius of the ParB-rich partition complex is 50 nm, which we estimated from our super-resolution images (Figure 1—figure supplement 3). In the model, the 50-nm ParB sphere is allowed to interact with infinite number of DNA-bound ParA-ATP dimers as long as there is an overlap between the ParB sphere and the ParA-ATP dimer sphere (the dimension of which was determined from the crystal structure of a ParA homolog (Leonard et al., EMBO J, 2005)). Multiple interactions between the ParB-rich partition complex and ParA dimers can be clearly seen in Video 3. We agree that this information was not clearly described in the main text of the original submission. The text has been revised to make it clearer.

Regarding the reviewer’s concern about the multiplicity of *parS* sites and their separation, it is important to note that there are only 2 *parS* sites in the *C. crescentus* chromosome and they are separated by only 42 nucleotides, which can be, at most, 14 nm long. Therefore, we believe that there is only one stable ParB/*parS* assembly complex per chromosome in *C. crescentus*. This is likely to be the case in most, if not all, plasmid systems.

However, the notion of separated ParB/*parS* assemblies remains interesting as it may be relevant to the *B. subtilis* chromosome case in which 10 *parS* sites are spread across a ∼2 Mb-nucleotide region flanking the origin of replication (Breier and Grossman, Mol Microbiol, 2007). By epifluorescence microscopy, this region appears as a single diffraction-limited spot (Lin et al., PNAS, 1997, Glaser et al., Genes Dev, 1997), indicating that the most distal *parS* sites are separated by ≤ 250 nm. As suggested by the reviewer, we have expanded the ‘diffusion-binding’ model to consider the cases of two ParB/*parS* assemblies (two spheres of 35 nm of radius) separated by a flexible linker of varying lengths to simulate the reviewer’s scenario. The reason for using a radius of 35 nm is to make the surface area of the two ParB spheres comparable to the surface area of the single partition complex sphere (50 nm of radius) used in the original model (2·π·35^2^ ≈ π·50^2^). Our simulations show that separating the *parS* sites by up to 400 nm had little, if any, positive effect on translocation (unpublished result), further supporting the notion that the elastic dynamics of the underlying DNA must be considered to obtain the translocation kinetics observed inside cells. Since the partition complex of *C. crescentus* is unlikely to form two or more ParB assemblies, we feel that these data do not belong to this manuscript. We would therefore prefer to publish these results elsewhere together with follow-up studies*.*